# Time-Aware World Model for Adaptive Prediction and Control

**Anh N. Nhu** [* 1]   **Sanghyun Son** [* 1]   **Ming Lin** [1]

## Abstract

In this work, we introduce the Time-Aware World Model (TAWM), a model-based approach that explicitly incorporates temporal dynamics. By conditioning on the time-step size, $\Delta t$, and training over a diverse range of $\Delta t$ values – rather than sampling at a fixed time-step – TAWM learns both high- and low-frequency task dynamics across diverse control problems. Grounded in the information-theoretic insight that the optimal sampling rate depends on a system's underlying dynamics, this time-aware formulation improves both performance and data efficiency. Empirical evaluations show that TAWM consistently outperforms conventional models across varying observation rates in a variety of control tasks, using the same number of training samples and iterations. Our code can be found online at: github.com/anh-nn01/Time-Aware-World-Model.

## 1. Introduction

Deep reinforcement learning (DRL) has recently demonstrated human-level or even expert-level capabilities on many highly complex and challenging problems, such as Go (Silver et al., 2016; 2017), Chess and Shogi games (Silver et al., 2018), and StarCraft II video game (Vinyals et al., 2019). DRL is also effectively adopted in a broad range of applications, including robotics (Wu et al., 2023; Koh et al., 2021; Johannink et al., 2019), autonomous vehicles (Kiran et al., 2021; Guan et al., 2024), and challenging control tasks where classical approaches fail to deliver satisfactory performance (Prasad et al., 2017; Nhu et al., 2023; Yang et al., 2015). Beyond model-free reinforcement learning (RL) methods, where an agent directly maps observation $o_t$ to action $a_t$ (Williams & Peng, 1989; Schulman et al., 2015; 2017; Haarnoja et al., 2018), there has been growing

---
[*]Equal contribution [1]Department of Computer Science, University of Maryland, College Park, United States. Correspondence to: Anh N. Nhu <anhu@umd.edu>, Sanghyun Son <shh1295@umd.edu>.

*Proceedings of the $42^{nd}$ International Conference on Machine Learning*, Vancouver, Canada. PMLR 267, 2025. Copyright 2025 by the author(s).

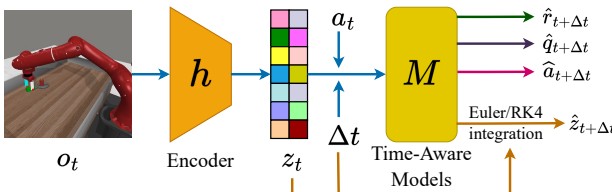

*Figure 1.* **Overall framework of our time-aware world model.** An encoder $h$ encodes the given observation $o_t$ into a latent vector $z_t$, which is then fed into various models with action $a_t$ and *time step size* $\Delta t$ to estimate values for action planning.

interest in model-based RL (MBRL). Unlike model-free approaches, MBRL constructs a model $\mathcal{M}$ that captures the underlying task dynamics (Sutton, 1990; Deisenroth & Rasmussen, 2011; Parmas et al., 2018; Kaiser et al., 2019; Janner et al., 2019), enabling the agent to plan actions using the learned *world model* $\mathcal{M}$. These model-based methods have gained traction over their model-free counterparts due to their superior sample efficiency and enhanced generalization capabilities (Ha & Schmidhuber, 2018; Hafner et al., 2019; 2020; 2023; Hansen et al., 2022; 2024).

Despite the remarkable success of world models in various control tasks, a critical factor in handling dynamical systems – the time step size, $\Delta t$ – has been largely overlooked in existing research. Specifically, the dynamics model $\mathcal{D} : (s_t, a_t) \rightarrow s_{t+1}$ , a fundamental component of the world model, governs state transitions from the current state $s_t$ to the next state $s_{t+1}$. Conventional approaches train $\mathcal{D}$ using experience tuples $(o_t, a_t, o_{t+1}, r_t)$ collected from interactions with the environment at a *fixed* time step size. However, this existing practice presents three key limitations (Thodoroff et al., 2022):

1. **Temporal resolution overfitting**: In current training pipelines, the observation time step $\Delta t$ are often fixed (e.g., $\Delta t = 2.5ms$ or frequency $f = 400Hz$). While smaller $\Delta t$ values stabilize simulations and prevent system aliasing, world models trained exclusively at a single, fixed $\Delta t$ often suffer significant performance degradation when deployed in real-world scenarios with different observation rates (e.g. $\Delta t = 20ms$ or $f = 50Hz$) due to compounding errors (Lambert et al., 2022; Wang et al., 2019) and computational costs of roll-out predictions. This discrepancy presents a

major challenge in extending the applicability of world models beyond simulated environments.

2. **Inaccurate system dynamics**: Training the dynamics model $\mathcal{M}$ on a fixed time step $\Delta t$ can lead to overfitting, as the model may not capture the true underlying dynamics of the task without conditioning on $\Delta t$. This can result in inaccurate state transitions and diminished generalization capabilities.

3. **Inefficient dynamics learning**: Although small $\Delta t$ values are essential for numerical stability and for capturing high-frequency dynamics, relying exclusively on fine-grained time steps results in inefficient sampling. Real-world systems often exhibit multiscale behavior, with some components evolving rapidly while others change more slowly. Sampling solely at a high frequency captures redundant information about the slower components, wasting computational resources and training data. This issue is exacerbated in systems with widely varying time scales, where no single sampling rate is optimal for every dynamical component.

Recent work ([Shaj Kumar et al., 2023](); [Lutter et al., 2021]()) has explored temporal aspects of world models, but primarily to improve long-horizon predictions through roll-outs of fixed-$\Delta t$ models. Our approach instead conditions the world model directly on $\Delta t$, enabling single-step predictions across arbitrary temporal intervals (Figure 1). Although previous methods can, in principle, accommodate variable time steps through repeated application, that strategy compounds errors and is computationally inefficient. By sampling strategically across multiple time scales, our method lets the model learn dynamics at different frequencies simultaneously, without increasing the number of training samples. Consequently, it generalizes effectively across varying observation rates – crucial for simulation-to-reality transfer – while maintaining superior sample efficiency compared with fixed-$\Delta t$ approaches.

In this work, We focus on the following question: *How can we efficiently train a world model $\mathcal{M}$ to capture the underlying task dynamics across varying time step sizes $\Delta t$ without increasing sample complexity?*

To answer this, we introduce a time-aware world model, $\mathcal{M}^{\text{TA}}$. Unlike earlier world models $\mathcal{M}$, our model conditions both the dynamics model and the reward function on $\Delta t$ (Figure 1), because these depend on the temporal gap between consecutive states. We construct $\mathcal{M}^{\text{TA}}$ by adapting the TD-MPC2 world model ([Hansen et al., 2024]()) and incorporating the fourth-order Runge–Kutta (RK4) method ([Butcher, 1987]()) to stabilize learning at large $\Delta t$ values (Section 4.1). We likewise modify the value model to accept $\Delta t$ as an additional input. Both models are trained on $\Delta t$ values log-uniformly sampled from a predefined interval.

Although one might expect $\mathcal{M}^{\text{TA}}$ to require more training samples than $\mathcal{M}$ because of the additional parameter $\Delta t$, this is not necessarily the case. According to the Nyquist–Shannon sampling theorem ([Shannon, 1949](); [Jerri, 1977]()), a signal with highest frequency f can be perfectly reconstructed by sampling at any frequency just above 2f. Thus, if the observation rate greatly exceeds 2f, the excess data are redundant – they contribute little to training the world model. A physical environment typically comprises multiple dynamical subsystems operating at different characteristic frequencies (Section 3.2.1). By training with a mixture of time-step sizes, we expose the model to a range of effective sampling frequencies, enabling each subsystem to be learned more efficiently (Section 3.2.3).

Inspired by the Nyquist–Shannon sampling theorem (Section 3.2.2), *we prove that our time-aware model trained on observation data sampled at a mixture of time steps achieves substantially better performance in learning world dynamics across different time steps at inference time, while using the same training budget as the baseline*. We demonstrate these results on a variety of control problems in Meta-World ([Yu et al., 2020]()) and PDE-control environments ([Zhang et al., 2024]()). Our contributions are summarized as follows:

1. We underscore the importance of a time-aware world model (TAWM) that conditions its dynamics modeling on the time step size $\Delta t$, a fundamental quantity in any dynamical system. By explicitly incorporating $\Delta t$, the model learns to capture the underlying task dynamics across a broad spectrum of step sizes. Our model's ability to generate a ***one-step prediction*** of the next state conditioned on $\Delta t$ *mitigates the well-known problem of compounding errors*. This approach is particularly suitable for real-world learning and control, where the observation rate may vary – or be substantially lower than – the default observation rate used for training.

2. Motivated by the Nyquist–Shannon sampling theorem and by the fact that real-world dynamics comprise many subsystems operating at different – and often unknown – frequencies, we propose a mixture-of-time-step training framework for TAWM that performs well across a range of observation rates at inference time, ***without increasing the number of training steps***. This perspective – training a world model with varying sampling rates – offers a new avenue for developing more efficient training strategies.

3. Empirically, we demonstrate that our time-aware world model can solve a range of control tasks at various observation rates ***without increasing either the amount of data or the number of training steps***. This capability helps narrow the gap between simulation environments and real-world control problems.

## 2. Related Work

Although model-free RL algorithms have gained popularity for their impressive performance, their inherent sample inefficiency limits their applicability to many real-world problems (Sutton et al., 1999; Williams & Peng, 1989; Barto et al., 1983; Schulman et al., 2015; 2017). Several works mitigate this limitation by integrating low-variance analytical gradients into the policy-learning process (Suh et al., 2022; Xu et al., 2022; Son et al., 2024).

In contrast, model-based RL (MBRL) tackles sample inefficiency at its root by training a dynamics model that *simulates* the environment, enabling agents to *predict* future states and action outcomes (Deisenroth et al., 2013). Because prediction is unconstrained by real-time sampling, MBRL is inherently more sample efficient. MBRL methods vary mainly in (1) how they define the world model $\mathcal{M}$ and (2) how they exploit $\mathcal{M}$ for training or planning. Historically, Gaussian processes (GPs) (Deisenroth & Rasmussen, 2011; Parmas et al., 2018) were widely used, but modern work favors neural networks (Ha & Schmidhuber, 2018; Hafner et al., 2019; 2020; 2023; Hansen et al., 2022; 2024) for their greater representational power. Among these, Dreamer and its variants (Hafner et al., 2019; 2020; 2023; Wu et al., 2023) train policies directly inside the learned model, whereas model-predictive-control (MPC) methods (Hansen et al., 2022; 2024) rely on action planning.

However, these approaches paid little attention to the temporal component when designing the world model $\mathcal{M}$, despite its importance in dynamical systems. To address its limitation in long-horizon prediction, Multi-Time-Scale World Models (MTS3) (Shaj Kumar et al., 2023) explicitly leverages temporal gaps when learning task dynamics. MTS3 captures state transitions over different prediction horizons $H$ but uses a single fixed $\Delta t$ during training; it therefore represents only two discrete time scales – *fast* and *slow* dynamics corresponding to $\Delta t$ and $H\Delta t$. In contrast, we incorporate a continuous-valued $\Delta t$ directly into the model, enabling single-step predictions across a range of inference step sizes. This allows our model to handle large temporal gaps (e.g., $\Delta t = 30ms$) that would otherwise require 12–20 smaller steps in a fixed-step model. Our approach is *architecture agnostic*: it builds on the TD-MPC2 framework (Hansen et al., 2024) yet crucially differs by explicitly conditioning the world model on $\Delta t$ and training on a mixture of step sizes, rather than assuming a fixed step throughout.

## 3. Background and Motivations

### 3.1. Model-Based Reinforcement Learning

We formulate the control problem as a Markov decision process (MDP) $\langle S, A, P, r, \gamma \rangle$, where $S$ is the state space, $A$ is the action space, $P : S \times A \times S \to \mathbb{R}$ is the (stochastic)

state-transition function, $r : S \times A \to \mathbb{R}$ is the reward function, and $\gamma \in [0, 1)$ is the discount factor. We refer to $P$ and $r$ as the *ground-truth* models, because in Section 4.1 we learn approximations of them within our world model.

The goal of reinforcement learning is to obtain a policy or planner $\pi$ that maximizes the expected discounted return along a trajectory $\tau = \{s_0, a_0, \ldots, s_{H-1}, a_{H-1}, s_H\}$ of length $H$:

$$\eta(\pi) = \mathbb{E}_{s_0, a_0, \ldots \sim \pi} \left[ \sum_{t=0}^{\infty} \gamma^t r(s_t, a_t) \right]. \qquad (1)$$

In model-based RL (MBRL), we train a world dynamics model consisting of a state-transition function $d_\phi : S \times A \to S$ and a reward function $r_\phi : S \times A \to \mathbb{R}$. The learned world model can then be exploited in multiple ways to derive a policy, for instance by using a planner such as TD-MPC2. Section 4.1.2 details our model formulation and the training pipeline for the time-aware world model.

### 3.2. Theoretical Motivations

Here we provide theoretical motivations to explain the sample efficiency of using a mixture of time step sizes during the training process of the time-aware world model.

#### 3.2.1. MULTI-SCALE DYNAMICAL SYSTEMS

In many control problems, the environmental dynamics can be decomposed into multiple subsystems, each evolving on a different temporal scale (Weinan, 2011). In other words, these subsystems can be described by distinct functions, each with its own highest frequency. Formally, consider a general dynamical system $\dot{x} = f(x, u, t)$, where $x$, $u$, and $t$ denote the state, control input, and time, respectively. Applying the Euler integration method, the next state $x'$ is

$$x' = x + f(x, u, t) \cdot \Delta t \qquad (2)$$

where $\Delta t$ is the time-step size. From a multi-scale perspective, this update can be rewritten as

$$x' = x + \sum_i f_i(x, u, t) \cdot \Delta t \qquad (3)$$

where $f_i(x, u, t)$ represents the dynamics of subsystem $i$. Each subsystem may evolve at its own temporal scale and therefore have its own highest frequency $f_{max}^i$.

#### 3.2.2. NYQUIST-SHANNON SAMPLING THEOREM

The Nyquist–Shannon sampling theorem states that a signal must be sampled at a rate of at least twice its highest frequency to avoid information loss (i.e., to prevent aliasing): $f_{sample} > 2f_{max}$ (Shannon, 1949). Here, $f_{sample}$ is the

observation rate, and $f_{max}$ is the highest frequency of the environment's dynamics in the MBRL context. If the observation rate is too low – such that $f_{sample} < 2f_{max}$ – we lose important dynamic details due to the large temporal gap $\Delta t$. This loss causes high-frequency components to fold back into lower frequencies, leading to inaccurate dynamics learned by the world model (Zeng et al., 2024). Although a higher $f_{sample}$ enables more accurate reconstruction of the environment's dynamics, an excessively high rate leads to oversampling, introducing redundant data that increase sample complexity and reduce learning efficiency. Therefore, choosing an appropriate observation rate $f_{sample}$ is crucial for balancing modeling accuracy and sample efficiency.

### 3.2.3. SIMULTANEOUSLY TRAINING ON MULTIPLE TEMPORAL RESOLUTIONS

To better align observation sampling with task dynamics across subsystems operating at different frequencies, we propose simultaneously training the world model at multiple temporal resolutions by varying the observation rate $f_{sample}$ (i.e., varying $\Delta t$) during training. As shown in Equation 3, the overall dynamics comprise several subsystems $f_i(\cdot)$, each with its own maximum frequency $f^i_{max}$. According to the Nyquist–Shannon sampling theorem, each subsystem is learned most efficiently at a distinct $f_{sample}$. By randomly varying $f_{sample}$ during training, we avoid undersampling high-frequency components and oversampling low-frequency ones, thereby training the subsystems $f_i(\cdot)$ more efficiently. Consequently, our time-aware world model learns the underlying task dynamics at multiple temporal resolutions without needing additional data (Section 5).

## 4. Methodology

In this section, we present our time-aware model formulation and training pipeline, designed to learn a world model that performs well across a range of observation rates. We introduce a novel time-aware training method that can be seamlessly integrated into any existing world-model architecture, enhancing robustness to observation-rate variations at inference time. We adopt TD-MPC2 (Hansen et al., 2024) as our baseline and adapt its architecture to train time-aware world models for various control tasks. We begin with a high-level overview of TD-MPC2 and its key architectural components. The overall framework is depicted in Figure 1.

### 4.1. Model Architecture

#### 4.1.1. BASELINE WORLD MODEL

TD-MPC2 (Hansen et al., 2024) is a model-based RL algorithm that captures task dynamics in a latent space – an "implicit" world model. Unlike reconstruction-based architectures (Ha & Schmidhuber, 2018; Hafner et al., 2023), TD-

MPC2 omits a decoder that maps latent representations back to raw observations. Recovering latent states in the high-dimensional observation space is computationally expensive and often unnecessary, as many elements are irrelevant to control. Once the latent dynamics are learned, TD-MPC2 rolls out latent predictions for local trajectory optimization with planning algorithms such as MPC (Hansen et al., 2022; 2024). The baseline TD-MPC2 model comprises five key components:

$$
\begin{aligned}
\text{Encoder:} \quad & z_t = h(o_t) \\
\text{Latent dynamic:} \quad & \hat{z}_{t+1} = D(z_t, a_t) \\
\text{Reward:} \quad & \hat{r}_t = R(z_t, a_t) \\
\text{Terminal value:} \quad & \hat{q}_t = Q(z_t, a_t) \\
\text{Policy prior:} \quad & \hat{a}_t = p(z_t)
\end{aligned}
$$

where $o_t$ is the raw observation at time step $t$, $z_t$ is its latent encoding, $\hat{z}_{t+1}$ is the predicted next latent state, $\hat{r}_t$ is the predicted immediate reward, $\hat{q}_t$ is the estimated $Q$-value of the current state–action pair, and $\hat{a}_t$ is the action sampled from the policy for the current state. At inference time, the Model Predictive Path Integral (MPPI) planner – an instance of Model Predictive Control (MPC) – is used for planning and action generation. All five component models are implemented using multilayer perceptrons (MLPs).

To train TD-MPC2, we maintain a replay buffer $\mathcal{B}$ that stores trajectories $(o_t, a_t, o_{t+1}, r_t)_{t=0}^{H}$ collected from the environment after each episode of length $H$. At the end of every episode, the model parameters are updated with batches randomly sampled from $\mathcal{B}$. The encoder $h$, dynamics model $D$, reward model $R$, and terminal-value model $Q$ are optimized jointly using a self-supervised consistency loss, a supervised reward loss, and a supervised temporal-difference loss for the terminal value. See Hansen et al. (2024) for details. The agent then resumes interaction with the environment, gathering additional data to augment $\mathcal{B}$ for subsequent training.

#### 4.1.2. TIME-AWARE WORLD MODEL

One notable limitation of TD-MPC2 and other state-of-the-art world models is that they ignore the effect of the observation rate $\Delta t$ at inference time. To introduce time awareness, we condition every component of the world model on $\Delta t$, as described below:

$$
\begin{aligned}
\text{Encoder:} \quad & z_t = h(o_t) \\
\text{Latent dynamics:} \quad & \hat{z}_{t+\Delta t} = z_t + d(z_t, a_t, \Delta t)\,\tau(\Delta t) \\
& \text{where } \tau(x) = \max(0,\ \log_{10} x + 5) \\
\text{Reward model:} \quad & \hat{r}_t = R(z_t, a_t, \Delta t) \\
\text{Value model:} \quad & \hat{q}_t = Q(z_t, a_t, \Delta t) \\
\text{Policy prior:} \quad & \hat{a}_t = p(z_t, \Delta t)
\end{aligned}
$$

Our time-aware model formulation is *architecture-agnostic* and can be readily integrated into any state-of-the-art world model. Because the observation encoder merely maps raw

observations to the latent space, without modeling dynamics, it is *not* conditioned on $\Delta t$. All other components that depend on the system's dynamics *are* conditioned on the time step size by taking $\Delta t$ as an explicit input.

While the baseline latent dynamics model $D$ maps a state–action pair $(z_t, a_t)$ directly to the next latent state $z_{t+1}$, we reformulate $D$ using Euler integration:

$$\hat{z}_{t+\Delta t} = D(z_t, a_t, \Delta t) = z_t + d(z_t, a_t, \Delta t) \cdot \tau(\Delta t),$$

where $d(\cdot)$ is implemented as an MLP. This formulation enforces the intrinsic property

$$z_{t+\Delta t}|_{\Delta t=0} = z_t, \ \forall z_t, a_t.$$

Thus, instead of learning the transition function $D$ directly, TAWM learns the latent-state derivative (gradient) function $d$, which is itself conditioned on $\Delta t$ to capture higher-order effects, and then advances the state by a single Euler step.

**State Transition**  Rather than integrating over the raw time step $\Delta t$, our dynamics model integrates using

$$\tau(\Delta t) = \max(0, \log_{10}(\Delta t) + 5).$$

The possible values of $\Delta t$ span several orders of magnitude – from a minimum of about $10^{-3}$ s to a maximum of roughly $5 \times 10^{-2}$ s – creating numerical challenges. Because latent vectors change only slightly between steps, the dynamics model $d(z_t, a_t, \Delta t)$ must scale appropriately across this wide range. Empirically, we observed convergence failures in some tasks (e.g., `mw-assembly`) when employing linear $\Delta t$ integration. By allowing the latent state to evolve with respect to the logarithm of the time step via $\tau(\Delta t)$, we effectively normalize $\Delta t$ to a narrower range, mitigating numerical issues and enabling more stable learning.

Before proceeding, we note that we experimented with two popular integration methods for our dynamics model: **(1)** Euler and **(2)** fourth-order Runge–Kutta (RK4), even though only Euler integration was introduced earlier. See Appendix A for RK4 implementation details. We include RK4 because it is broadly applicable to both simple linear and complex nonlinear systems and is standard in physical simulation. The integration method thus serves as a tunable hyperparameter for maximizing TAWM's performance and learning efficiency across control tasks.

For most Meta-World robot-manipulation tasks, the dynamics are simple enough to be well approximated by Euler integration. Our ablation study shows that TAWM with Euler integration outperforms its RK4 counterpart on the majority of Meta-World tasks, indicating that Euler suffices in these settings (Figure 3). By contrast, RK4's benefits become more pronounced for tasks with complex dynamics, such as those in PDE-control environments (Figure 3).

---

**Algorithm 1** Time-Aware World Model Training Paradigm

1: Initialize task environment $\mathcal{E}$
    Initialize time-aware world model $\mathcal{M}^{TA}$
2: Set experience buffer $\mathcal{B} \leftarrow \emptyset$
3: **repeat**
4:   **for** each *episode* **do**
5:     Set $\Delta t \sim$ Log-Uniform$(\Delta t_{min}, \Delta t_{max})$
        ▷ Meta-World: $\Delta t_{min} = 0.001$, $\Delta t_{max} = 0.05$
                ($\Delta t_{default} = 0.0025$)
        ▷ Can be either Log-Uniform or Uniform
6:     Set $step \leftarrow 0$
7:     **while** $step <$ Horizon $H$ **do**
8:       $a_t \leftarrow \mathcal{M}^{TA}.\mathtt{act}\,(o_t, \Delta t)$
9:       Execute $a_t$ in $\mathcal{E}$, get back $(o_{t+\Delta t}, r_t)$
10:      Add transition $(o_t, a_t, o_{t+\Delta t}, r_t, \Delta t)$ to $\mathcal{B}$
11:      $\{(o_t, a_t, r_t, o_{t+\Delta t}, \Delta t)_{1:B}\} \sim \mathcal{B}$: update time-aware world model $\mathcal{M}^{TA}$
12:      $step \leftarrow step + 1$
13:    **end while**
14:  **end for**
15: **until** reach $N$ training steps
16: **return** $\mathcal{M}^{TA}$

---

### 4.2. Training Procedure Using a Mixture of $\Delta t$

Algorithm 1 summarizes our training procedure, in which we vary the observation rate to encourage the model to learn the underlying dynamics at multiple temporal resolutions. The world model acquires a spatiotemporal representation of the environment by sampling observations from the various dynamical subsystems at specific points in time and space. According to the Nyquist–Shannon sampling theorem, a signal must be sampled at a rate of at least $1/(2\mathrm{f}_{max})$, where $\mathrm{f}_{max}$ is the highest frequency present in the band-limited signal. Because there is no systematic way to determine $\mathrm{f}_{max}$ for every subsystem, we instead sample observations at several temporal rates to capture the underlying dynamics more effectively.

At the start of each episode (Algorithm 1), we sample $\Delta t$ from a log-uniform distribution and set the observation/control rate to $1/\Delta t$. A log-uniform distribution assigns equal probability mass to every octave of time scales, yielding good coverage across multiple orders of magnitude. This is especially useful when $\Delta t_{\max} \geq \frac{1}{2\mathrm{f}_{max}}$, where $\mathrm{f}_{max}$ is the highest frequency of the task dynamics. For tasks with dynamics slow enough to be captured by $\Delta t_{\max}$, however, uniform sampling may outperform log-uniform sampling, particularly at larger $\Delta t$ values (Figure 7).

### 4.3. Theoretical Analysis of TAWM's Sample Efficiency

We present theoretical insights into TAWM's sample efficiency, showing that it can be trained to capture task dynam-

ics across temporal scales *without increasing the number of samples or training steps* and *without sacrificing performance*. We begin with two core assumptions:

1. **Model capability**: Because our time-aware training framework is model-agnostic, it inherits the limitations of the underlying world-model architecture. We assume that this architecture has enough representation power to capture the system's complex dynamics.

2. **Sampling frequency**: We focus our analysis on those subsystems and time intervals for which $\Delta t_{\max}$ can effectively capture the underlying dynamics. We assume that the empirically determined range $[\Delta t_{\min} = 0.001s, \Delta t_{\max} = 1s]$ adequately captures the behavior of most subsystems.

In the following discussion, we denote by $\bar{f}$ the ground-truth dynamics function, which satisfies the equation:

$$z_{t+\Delta t} = z_t + \bar{f}(z_t, a_t, \Delta t) \cdot \Delta t.$$

Also, we say that an environment's dynamics can be *fully captured* with a time step $\Delta \bar{t}$ if its dynamics function $\bar{f}$ satisfies:

$$||\bar{f}(z_t, a_t, \Delta \bar{t}) - \bar{f}(z_t, a_t, \Delta t)|| < \epsilon, \ \forall \Delta t < \Delta \bar{t},$$

where $\epsilon \ll 1$ is a sufficiently small constant.

Then, the following lemma shows that the optimal time-aware dynamics function $d^*$ (Section 4.1) can approximate dynamics for all $\Delta t < \Delta \bar{t}$ through simple interpolation.

**Lemma 4.1.** *When the environment's dynamics can be fully captured with $\Delta \bar{t}$, the optimal dynamics function $d^*$ satisfies the following relation for any smaller time step $\Delta t < \Delta \bar{t}$:*

$$||d^*(z_t, a_t, \Delta t) - d^*(z_t, a_t, \Delta \bar{t}) \cdot \frac{\Delta t}{\tau(\Delta t)} \cdot \frac{\tau(\Delta \bar{t})}{\Delta \bar{t}}|| < \epsilon.$$

**Proof.** Please see Appendix H.1.

Note that the interpolation factor $\frac{\Delta t}{\tau(\Delta t)} \cdot \frac{\tau(\Delta \bar{t})}{\Delta \bar{t}}$ is common to every training sample; regardless of $z_t$ or $a_t$, this relationship holds. Based on this observation, we introduce our third core assumption as follows.

3. **Interpolation Learning**: During training, the dynamics function $d$ effectively learns the relationship described in Lemma 4.1 as the interpolation factor is shared across all samples.

With this assumption, we can prove the following lemma, which states that error reductions at larger temporal scales contribute to reductions at smaller scales during training.

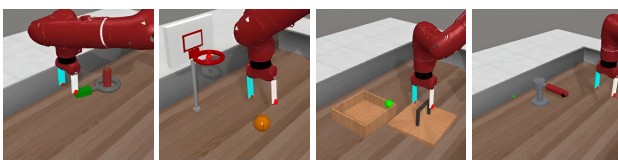

*Figure 2.* **Visualizations of the Meta-World control tasks.** From left to right: *Assembly, Basketball, Box Close, Faucet Open.* The default time step size of all environments is $\Delta t = 2.5ms$.

**Lemma 4.2.** *When the environment's dynamics can be fully captured with $\Delta \bar{t}$, reducing the modeling error at $\Delta \bar{t}$ lowers the upper bound of the error for every $\Delta t < \Delta \bar{t}$.*

**Proof:** Please see Appendix H.2.

Building on this lemma, we conjecture that improvements gained at one temporal scale transfer to all smaller scales, thereby enhancing TAWM's sample efficiency. In the next section, we empirically confirm this conjecture by demonstrating TAWM's high sample efficiency.

## 5. Experiments

We address three key questions in our experiments: **(1)** Under the same planner, does TAWM match the baseline's performance at the default observation rate while avoiding degradation at lower rates? **(2)** At which observation rates does TAWM outperform the baseline? **(3)** Does TAWM require more training data than the baseline? We primarily investigate these questions in Section 5.1 and present ablation studies on sampling strategies in Section 5.2.

### 5.1. Evaluations on Control Problems

To answer our main questions and evaluate the performance and learning efficiency of our time-aware world model, we conducted experiments on several control tasks from the Meta-World simulation suite, which uses a default time step of $\Delta t = 2.5ms$ $(0.0025s)$ (Yu et al., 2020). We tested *nine* diverse tasks, each with distinct goals and motion characteristics: Assembly, Basketball, Box Close, Faucet Open, Hammer, Handle Pull, Lever Pull, Pick Out of Hole, and Sweep Into. Representative environments are shown in Figure 2, and Appendix B provides renderings of all nine tasks. Following Hansen et al. (2024), we report the success rate (%) as the primary metric for comparing the time-aware and baseline models on these Meta-World control tasks.

On top of that, our experiments also include three non-linear, one-dimensional PDE-control tasks from `control-gym` (Zhang et al., 2024): Burgers, Allen–Cahn, and Wave. For these tasks, we use the (continuous) total episode reward as the primary metric. Because of space constraints, we defer detailed problem descriptions, visualizations, and additional analysis of these environments to Appendix G.

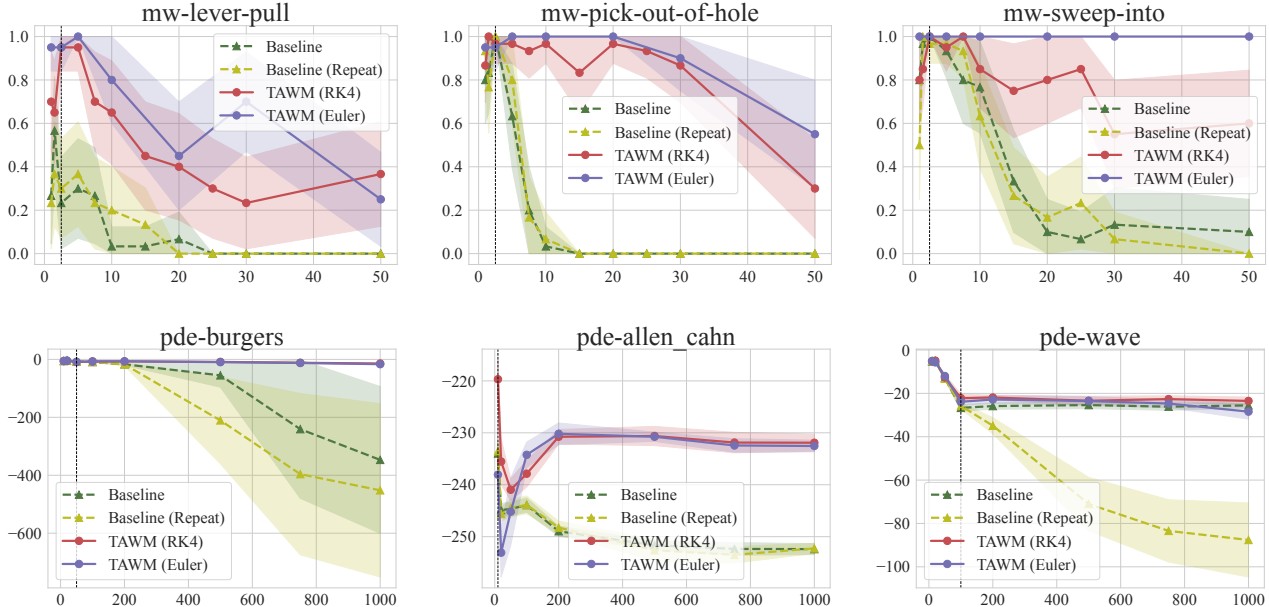

*Figure 3.* **Performance comparisons for 3 tasks from Meta-World (Up), and 3 tasks from PDE-Control (Down)**. The x-axis corresponds to the evaluation $\Delta t$ (in $ms$), and the y-axis corresponds to success rate (Meta-world) and episode reward (PDE-Control). We trained our TAWM model using either RK4 or Euler integration method with log-uniform sampling strategy. The baseline method was trained using only default time step ($\Delta t = 2.5\,ms$ for Meta-World tasks; $\Delta t = 50\,ms/10\,ms/100\,ms$ for PDE-Burgers, PDE-Allen-Cahn, PDE-Wave, respectively), which is signified with the black dashed line in each environment. For fair comparisons, we extended the baseline method by repeating the same actions for the larger evaluation time steps than the default one. For instance, when the evaluation $\Delta t = 5ms$, we repeat the same action for $5/2.5 = 2$ times. Plots show mean and 95% confidence intervals over 3 seeds, with 10 evaluation episodes per seed.

**Training Setup.** We built our time-aware model on the base TD-MPC2 architecture, using the same default training hyperparameters (e.g., model size, learning rate, and horizon). As shown in Algorithm 1, we randomly varied $\Delta t$ during training to create a mixture of training observation rates. Because there is no systematic way to identify the highest frequency in each task's dynamics, we set $\Delta t$ to the range $[0.001, 0.05]$ for Meta-World tasks and $[0.01, 1.0]$ for PDE-control tasks. For Meta-World tasks, each TAWM was trained for 1.5 million steps, which required roughly 40–45 hours on a single NVIDIA RTX 4000 GPU (16 GB VRAM) and 32 CPU cores. For PDE-Allen-Cahn and PDE-Wave, the TAWM and baseline models were trained for 1M steps. For PDE-Burgers, all models were trained for 750k steps.

**Performance Comparisons Across Different $\Delta t$.** To assess the time-aware model's performance at different inference-time observation rates, we evaluated it on multiple tasks with varying $\Delta t$. As shown in Figures 3 and 11, both TAWM variants – one using RK4 integration and the other using Euler integration – outperform the baseline (trained at a fixed $\Delta t = 2.5$ ms) across all tasks under identical hyperparameters. We observed similar gains in every other Meta-World task (Appendix E), indicating that the time-aware model effectively learns both fast and slow dynamics without increasing sample complexity. Empirically, the dy-

namics of some Meta-World tasks are sufficiently simple to be well approximated by TAWM with Euler integration, leading to better performance than with RK4 integration.

**Effects of using Mixtures of $\Delta t$.** To demonstrate the effectiveness of training the world model at multiple temporal resolutions $\Delta t$, we compare our approach with baselines trained only at fixed $\Delta t$ values different from the default $\Delta t = 2.5ms$. Figure 4 shows that our time-aware model outperforms all baselines across three Meta-World tasks. Most notably, when baselines are trained solely at low observation rates (e.g., $\Delta t \geq 10ms$), they fail to converge and achieve zero success on every task at every evaluation rate. In contrast, by training with a mixture of time steps, our time-aware model consistently surpasses any baseline trained at a single step size, regardless of that fixed $\Delta t$. These results suggest that environmental dynamics comprise multiple subsystems, each describable as a time-dependent, spatially parameterized function with its own highest frequency. Varying the observation rate (i.e., varying $\Delta t$), therefore enables the world model to learn these underlying subsystems more effectively.

**Convergence Rate on Various $\Delta t$.** To evaluate the sample efficiency of TAWM, we compare the episode success rate curves across different control tasks between our

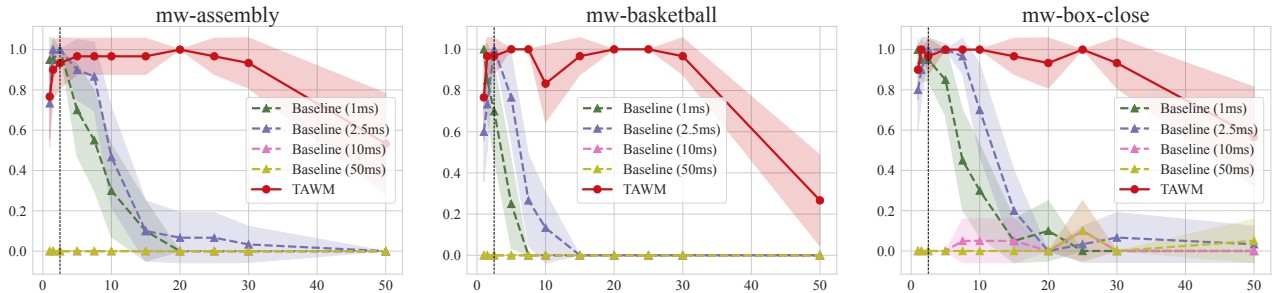

*Figure 4.* **Performance comparisons to baseline methods trained on different $\Delta t$ values for Meta-World tasks**. The x-axis corresponds to the evaluation $\Delta t$, and the y-axis corresponds to the success rate. The non–time-aware baseline models were trained with different fixed $\Delta t$ values (shown in the legend). Our TAWM model employs RK4 integration and log-uniform $\Delta t$ sampling. TAWM *outperforms all baseline models trained with fixed time-step sizes*. When the baselines are trained only at low observation rates ($\Delta t \geq 10$ms), they fail on *all* tasks. The dashed black line indicates the default time-step size ($\Delta t = 2.5$ms). Means and 95% confidence intervals are computed over 3 seeds, each evaluated for 10 episodes.

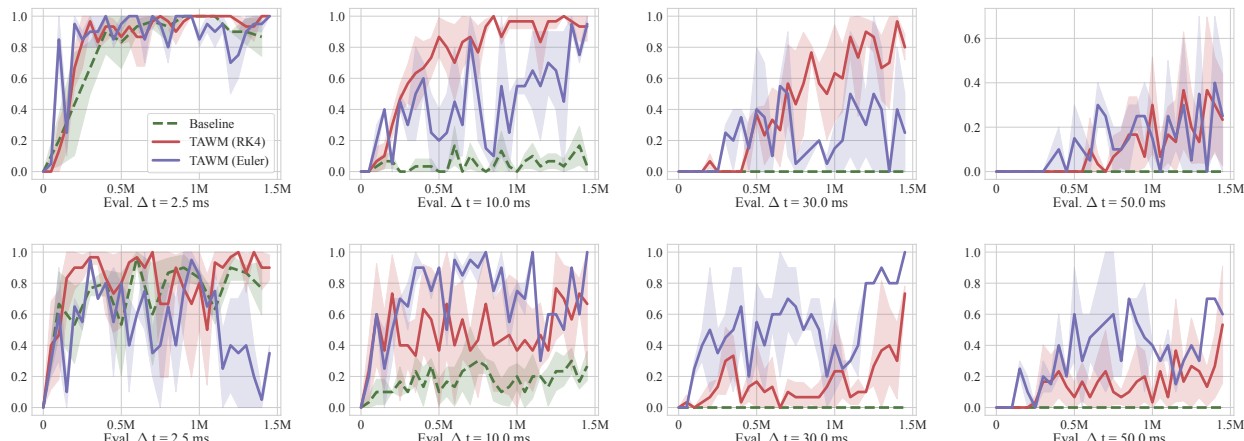

*Figure 5.* **Learning curve comparison on Meta-World Basketball (Up) and Meta-World Lever-Pull (Down)**. The x-axis corresponds to the evaluation $\Delta t$ (in $ms$), and the y-axis corresponds to the success rate. We trained our TAWM using either RK4 or Euler integration method with log-uniform sampling strategy. The baseline method was trained using only the default time step ($\Delta t = 2.5ms$). Plots show mean and 95% confidence intervals over 3 seeds, with 10 evaluation episodes per seed.

TAWM models and non-time-aware baseline models trained with a fixed default time step of $\Delta t = 2.5$ ms. We assess the success rate curves of TAWM and baseline models under different evaluation $\Delta t$ values across various control tasks throughout training. In Figure 5, we present the learning curves for the Meta-World Basketball and Lever-Pull tasks. Despite having to learn state transitions under varying time step sizes, our TAWM with RK4 integration method converges at least as quickly as the baseline when evaluated at the default $\Delta t = 2.5$ ms – the exact time step on which the baseline was specifically trained. Compared to the non-time-aware baseline specialized at only $\Delta t = 2.5$ ms, TAWM (RK4) still converges to a higher success rate at $\Delta t = 2.5$ ms after 1.5M training steps. Both TAWM variants (RK4 and Euler) reach 100% success on the Meta-World Basketball task at $\Delta t = 2.5$ ms, outperforming the baseline.

When evaluated at inference $\Delta t$ values greater than the de-

fault ($\Delta t > 2.5$ ms), both TAWM (RK4) and TAWM (Euler) significantly outperform the non-time-aware baselines. We refer the readers to Figure 12 and Figure 13 in Appendix F, as well as Figure 17 in Appendix G, for additional learning curves on other control tasks. *These results demonstrate that, despite having to learn task dynamics across different temporal scales,* ***our time-aware model does not require additional training steps or samples to converge to a sufficiently accurate model capable of effectively solving control tasks at varying observation rates***.

**Comparison to MTS3** Finally, we compare our model's performance with the Multi Time Scale World Model (MTS3) (Shaj Kumar et al., 2023), a closely related approach that shares the high-level motivation of modeling world dynamics across multiple temporal scales, as introduced in Section 2. We would like to refer the readers to Appendix D for detailed descriptions of MTS3's experi-

mental settings. In Figure 6, we observe that our TAWM consistently outperforms MTS3 across various settings of MTS3's slow dynamics hyperparameter $H$. Notably, MTS3 exhibits rapid performance degradation as the evaluation time step increases, suggesting that it suffers from compounding errors due to long-horizon predictions. In contrast, our TAWM shows greater robustness at lower observation rates, especially on the Meta-World Faucet-Open task in which TAWM's success rate remains at $\approx 90\%$ at $\Delta t = 50$ ms (0.05 s).

## 5.2. Ablation studies on sampling strategies

The $\Delta t$ sampling strategy is a tunable hyperparameter to optimize TAWM's performance and efficiency. In this section, we conduct an ablation study on the impact of the $\Delta t$ sampling strategy during training on the performance and sample efficiency of TAWM. In Meta-World tasks, we trained the same TAWM architecture in the same $\Delta t$ range using two different sampling strategies: **(1)** Log-Uniform$(0.001\,s, 0.05\,s)$ s and **(2)** Uniform$(0.001\,s, 0.05\,s)$. In some tasks, such as Meta-World Assembly and Basketball, uniform sampling may outperform log-uniform sampling – particularly when the task dynamics are sufficiently slow to be captured by $\Delta t_{\max}$.

Figure 7 shows that while TAWM trained with the uniform sampling strategy generally performs better in most environments and achieves higher performance at low sampling rates ($\Delta t \geq 30$ ms), it exhibits lower success rates at smaller inference $\Delta t$ in some environments, such as Meta-World Assembly. Additional results in other environments are presented in Figure 8, which show a similar pattern. Regardless of the $\Delta t$ sampling strategy, time-aware models consistently outperform non-time-aware baselines. Therefore, depending on the task dynamics, ***TAWM can be effectively and efficiently trained with any reasonable $\Delta t$ sampling strategy and is not restricted to log-uniform or uniform schemes***.

## 6. Conclusion

In this paper, we introduce a novel *time-aware world model (TAWM)* that adaptively learns task dynamics across different temporal scales. By explicitly conditioning the dynamic model on the time step size $\Delta t$ and training it with a mixture of temporal scales (via $\Delta t$ sampling), TAWM achieves robust performance across varying observation rates on diverse control tasks *without requiring additional training steps or samples. Empirical results show that our model consistently outperforms the non-time-aware baseline models*, which are trained only on a fixed time step size $\Delta t$ for different training $\Delta t$ values. We hope that the insights and results in this paper offer a new perspective on world model training, thereby contributing to the community a new, efficient, yet simple world model training method.

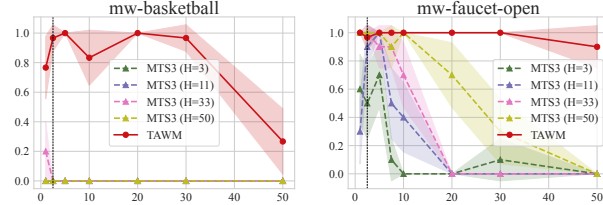

*Figure 6.* **Performance comparisons to MTS3 across different evaluation $\Delta t$'s on Meta-World tasks**. The x-axis corresponds to the evaluation $\Delta t$, and the y-axis corresponds to the success rate. The MTS3 models were trained under different slow dynamics settings, indicated by the $H$ values shown in the legend, using offline data with 4M transitions. Our TAWM employs RK4 integration method with log-uniform sampling strategy and was trained for 1.5M steps. The dashed black line indicates the default time-step size ($\Delta t = 2.5$ms). Means and 95% confidence intervals are computed over 3 seeds for TAWM, each evaluated for 10 episodes. TAWM outperforms MTS3 across different $H$ settings, especially at large $\Delta t$ values.

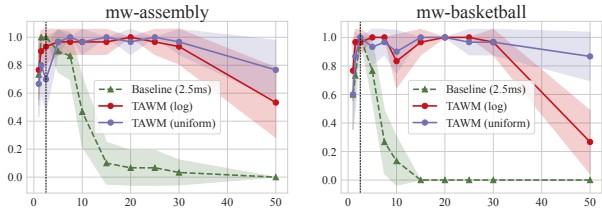

*Figure 7.* **Ablation Study on $\Delta t$ Sampling Strategy**. We trained our TAWM models with log-uniform and uniform sampling strategies, both of which use the RK4 integration method. The baseline method was trained using only default time step ($\Delta t = 2.5$ms), which is signified with the black dashed lines. The x-axis corresponds to the evaluation $\Delta t$, and the y-axis corresponds to the success rate. Means and 95% confidence intervals are computed over 3 seeds, each evaluated for 10 episodes.

**Limitations and Future Work.** We have yet to develop a systematic methodology to analytically compute the highest frequency of the underlying task dynamics; therefore, the upper bound for the training time step size $\Delta t_{\max}$ is determined empirically. This limitation necessitates a search for $\Delta t_{\max}$ when applying TAWM to new environments (e.g., autonomous driving or PDE controllers) to improve training efficiency and convergence. An interesting direction for future work is to develop an automatic method to identify the highest frequency of task dynamics, which would determine the lower bound for sampling frequency (or the upper bound for $\Delta t_{\max}$) in training TAWM. We also plan to extend our current deterministic dynamics model to a probabilistic one to better capture real-world state transitions, particularly at larger $\Delta t$ values and/or over longer horizons.

## Acknowledgments

This research is supported in part by the National Science Foundation and the U.S.Army Research Labs Cooperative Agreement on "*AI and Autonomy for Multi-Agent Systems*".

## Impact Statement

This paper presents a novel, model-agnostic approach to efficiently train Time-Aware World Models (TAWM) capable of handling varying observation rates and control frequencies without increasing sample complexity. Our method enables: **(1)** training a single world model usable across multiple observation rates and control frequencies, **(2)** avoiding the need to retrain a different model for each frequency, and **(3)** significantly reducing computational and energy consumption. This training method is model-agnostic, is compatible with any existing world model architecture, and demonstrates the feasibility of time-aware training without increasing training time or data. This work promotes efficient, green computing in academic and industrial research of world models. Another potential societal impact includes applications in autonomous military systems. The authors do not foresee any other immediate consequences resulting from this work.

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

# Appendix

## A. Descriptions of 4th-order Runge-Kutta integration.

In this section, we extend our description of the 4th-order Runge-Kutta (RK4) integration mentioned in Section 4.1.2. The detailed RK4 integration is as follows:

$$
\begin{aligned}
k_1 &= d(z_t, a_t, \Delta t) \\
\hat{z}_1 &= z_t + d\left(z_t, a_t, \Delta t/2\right) \cdot \tau\left(\Delta t/2\right) \\
k_2 &= d\left(\hat{z}_1, a_t, \Delta t\right) \\
\hat{z}_2 &= z_t + d\left(\hat{z}_1, a_t, \Delta t/2\right) \cdot \tau\left(\Delta t/2\right) \\
k_3 &= d\left(\hat{z}_2, a_t, \Delta t\right) \\
\hat{z}_3 &= z_t + d\left(\hat{z}_2, a_t, \Delta t\right) \cdot \tau(\Delta t) \\
k_4 &= d\left(\hat{z}_3, a_t, \Delta t\right) \\
\hat{z}_{t+\Delta t} &= z_t + \frac{1}{6}(k_1 + 2k_2 + 2k_3 + k4) \cdot \tau(\Delta t)
\end{aligned}
\tag{4}
$$

Consistent with the notations in Section 4.1.2, $z_t, a_t$ denotes the latent state-action pairs at time $t$, $d(\cdot)$ denotes our dynamic model parameterized by a neural network, and $\hat{z}_i$ ($i \in 1, 2, 3$) are the intermediate middle points. The final prediction of next latent state under time step size $\Delta t$ is $\hat{z}_{t+\Delta t}$.

## B. Meta-World Task Dynamics Visualizations.

To illustrate the diversity and key differences in dynamics across control tasks in our experiments, we present additional visualizations of different Meta-World tasks. Figure 9 and Figure 10 show sequences of renderings from the task initialization to the task completion (from left to right) of different control tasks.

## C. Additional results for ablation study of $\Delta t$ sampling strategies.

In this section, we provide additional results for the ablation study on the impact of $\Delta t$ sampling strategy on the TAWM's performance and efficiency. Specifically, we trained the same TAWM over the same range of $\Delta t$ values using two different sampling strategies: **(1)** Log-Uniform(1 ms, 50 ms) and **(2)** Uniform(1 ms, 50 ms). The performance of TAWMs trained with different $\Delta t$ sampling strategies, along with the non-time-aware baselines, is shown in Figure 8.

Figure 8 shows that while our time-aware models trained with the uniform sampling strategy generally perform better across most environments – and significantly outperform at low sampling rates (inference $\Delta t \geq 30$ ms) – they exhibit lower success rates at smaller inference $\Delta t$ in certain environments, such as Meta-World Lever-Pull. Regardless of the $\Delta t$ sampling strategy, TAWMs consistently outperform the non-time-aware baselines. Therefore, **our time-aware model can be efficiently and effectively trained with any reasonable sampling strategy and is not restricted to either log-uniform or uniform sampling**.

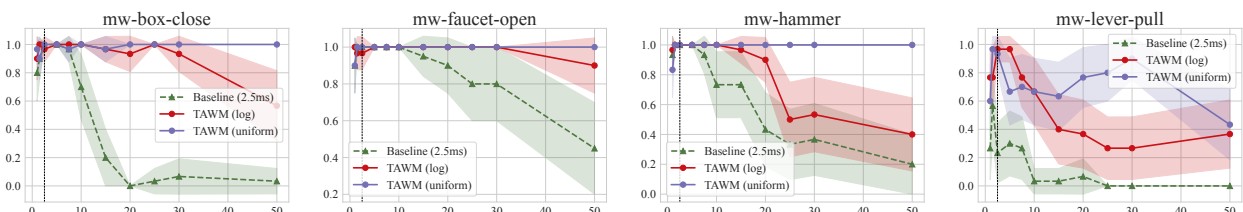

*Figure 8.* **Ablation Study on $\Delta t$ Sampling Strategy**. We trained our TAWM models with log-uniform and uniform sampling strategies, both of which use the RK4 integration method. The baseline method was trained using only default time step ($\Delta t = 2.5ms$), which is signified with the black dashed lines. The x-axis corresponds to the evaluation $\Delta t$, and the y-axis corresponds to the success rate. Means and 95% confidence intervals are computed over 3 seeds, each evaluated for 10 episodes.

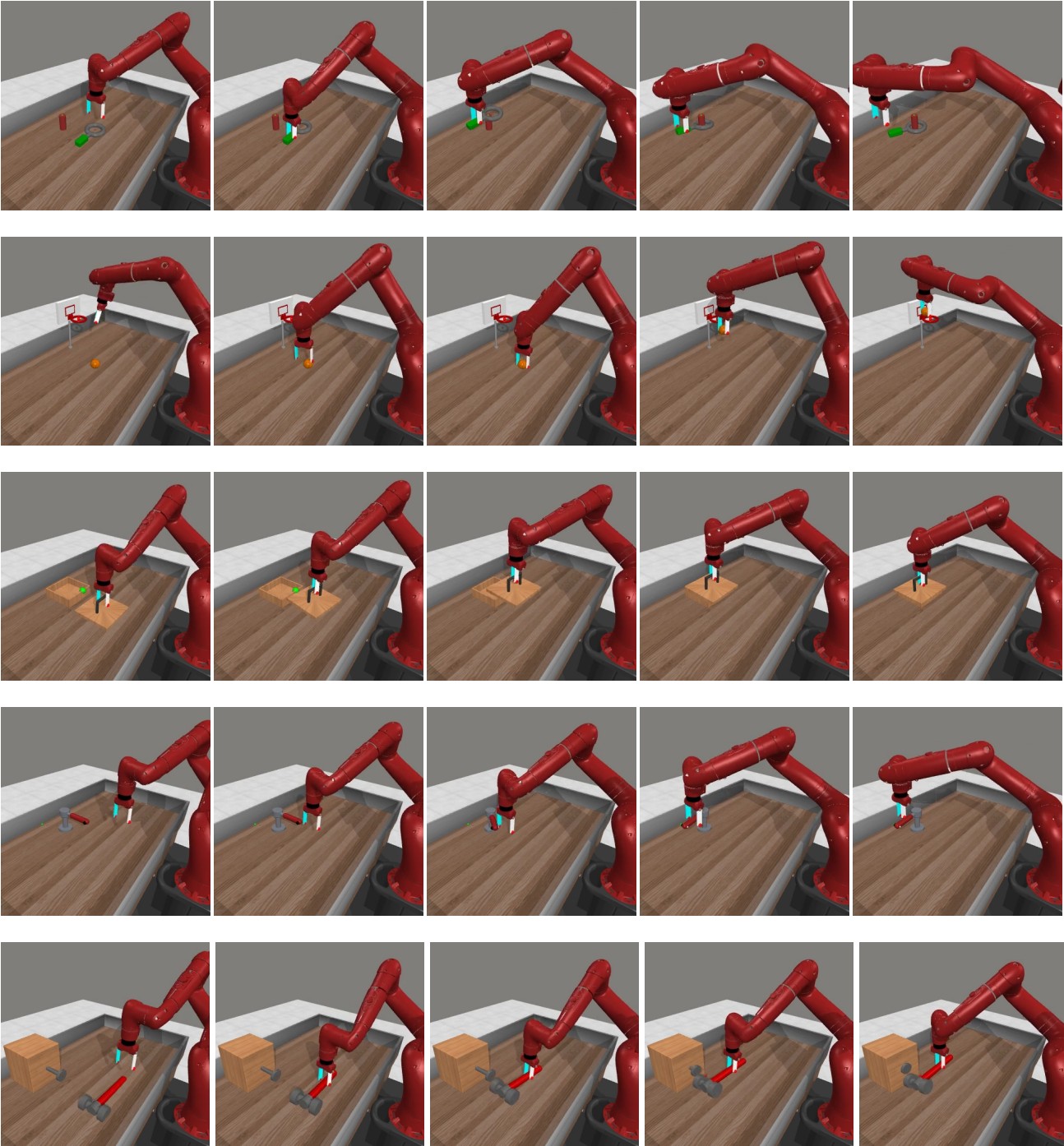

*Figure 9.* **Visualizations of the Meta-World control tasks.** From top to bottom: *(1) Assembly, (2) Basketball, (3) Box Close, (4) Faucet Open, (5) Hammer.* From left to right: sequences of renderings from task initialization to completion.

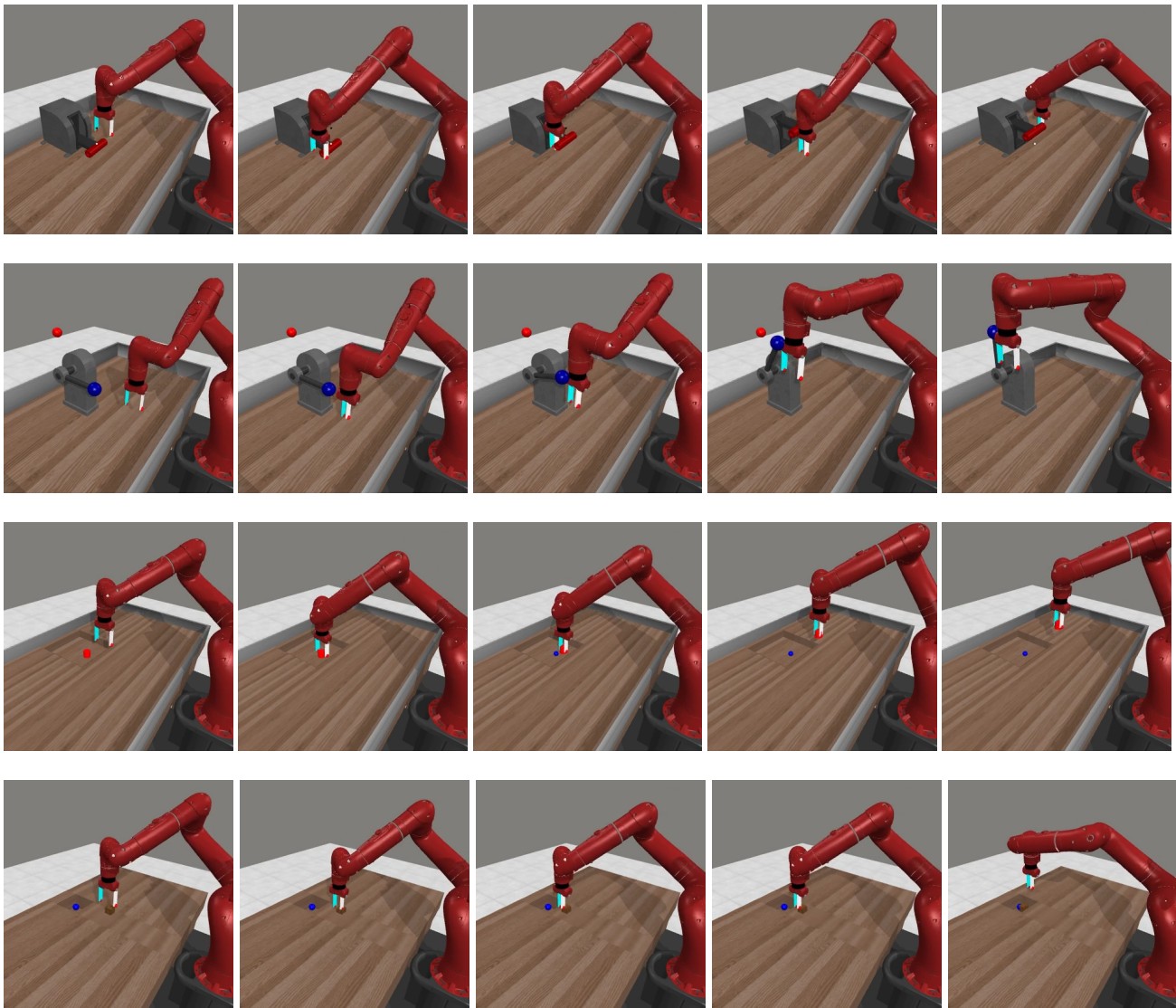

Figure 10. **Visualizations of the Meta-World control tasks.** From top to bottom: *(6) Handle Pull, (7) Lever Pull, (8) Pick Out Of Hole, (9) Sweep Into*. From left to right: sequences of renderings from task initialization to completion.

Developing a more adaptive and optimal $\Delta t$ sampling strategy tailored to specific objectives and scenarios is a promising direction for future work – particularly to achieve strong performance across both small and large $\Delta t$ values and to accelerate convergence rates. In the meantime, TAWM with either log-uniform or uniform sampling strategy has demonstrated practical effectiveness, as shown by our experimental results.

## D. Comparisons with Multi Time Scale World Model (MTS3).

As mentioned in Section 2, Multi Time Scale World Model (MTS3) (Shaj Kumar et al., 2023) is a closely related work sharing the high-level motivation with our work: to model the world dynamics at multiple temporal levels. Specifically, MTS3 proposes a probabilistic approach to jointly learn the world dynamics at two temporal abstractions: task level (slow dynamics/timescale) and state level (fast dynamics/timescale). These 2 timescales are separately learned by two state space models (SSMs): $SSM^{fast}$ and $SSM^{slow}$, where $SSM^{fast}$ learns the dynamics evolving at original small timestep $\Delta t$ of the dynamical systems and $SSM^{fast}$ learn the slow dynamics evolving at $H\Delta t$. Although this approach also explicitly considered different temporal abstraction levels in learning the world dynamics, there are several critical differences between MTS3 compared to our work:

1. **Models vs Training method:** (Shaj Kumar et al., 2023) proposes a model architecture to learn a world model with several discrete temporal abstraction levels. On the other hand, we proposed a *simple yet effective and efficient time-aware training method* that can be employed to train any world model architecture.

2. **Discrete vs continous timescales:** The original MTS3 currently only handle only 2 timescales: $\Delta t$ and $H\Delta t$, where both $\Delta t$ and $H$ is fixed in the training process. Although the MTS3 can be adapted to learn multiple timescales, the number of timescales is limited to a discrete value. Furthermore, the $SSM^{slow}$ (slow dynamic model) does not directly model state transition under large temporal gap $\Delta t$ (or low observation rate) but rather learns the task latents to guide $SSM^{fast}$ to long-horizon predictions. On the other hand, our time-aware approach can directly predict the future states $s_{t+\Delta t}$ under large $\Delta t$.

3. **Multi-step vs one-step prediction:** MTS3 considers the future prediction under large $\Delta t$ as a long-horizon prediction problem. Specifically, to predict $s_{t+\Delta t}$, MTS3 discretizes the long temporal gap into several smaller timesteps: $\Delta t = M\Delta t_{fast}$, where $\Delta t_{fast}$ is the original timestep size $SSM^{fast}$ is trained with and $M \in \mathbb{N}^{+}$. MTS3 then iteratively applies the model $M$ times to predict $s_{t+\Delta t}$. This timestep discretization approach has 2 critical limitations: **(1)** MTS3 cannot model state transitions under $\Delta t$ that is not divisible by $\Delta t_{fast}$ and **(2)** multi-step predictions are vulnerable to compounding errors, a well-known problem in long-horizon modeling. On the other hand, our model can directly predict the next state with a *one-step prediction*, effectively alleviating the compounding error problem.

4. **Inference efficiency:** Another disadvantage of multi-step prediction is inference inefficiency. In contrast, our time-aware model can efficiently predict long-term future states without sacrificing computational efficiency by using one-step prediction.

5. **Prediction-only vs Control:** As acknowledged by (Shaj Kumar et al., 2023), the original MTS3 is strictly a prediction model. On the contrary, our time-aware model can be used efficiently with a planner to solve control problems.

We conduct empirical comparisons between MTS3 and our proposed time-aware model on the control problem, extending beyond the prediction-only scope in MTS3. First, MTS3 is trained with offline data consisting of $4 \times 10^6$ (4M transitions) collected from random trajectories (10%), partially-trained policy's trajectories (20%), and fully-trained expert policy trajectories (80%). Since MTS3 is strictly prediction-focused and is not designed for controls, we combined MTS3 with MPPI planners and our world model's trained value and reward functions. Implementation-wise, we replaced our dynamic model with MTS3 and kept all other components unchanged, including the planner (MPPI), learned value function, and learned reward function. This design ensures a fair comparison between the models, as any performance gap is attributed solely to the difference between MTS3 and our dynamic model.

We kept the default hyperparameter settings as in the original MTS3 paper and codebase while varying the slow dynamics $H$ to investigate the impacts of $H$ on MTS3's performance. The authors of MTS3 suggested that using $H = \sqrt{T}$, which is $\sqrt{99} \approx 10$ in our experiments on Meta-World environments. We chose $H = 11$ to divide the episodes into equal-length local SSM windows. The MTS3 inference stepping are adjusted such that when evaluated on $\Delta t_{eval} > \Delta t_{train}$, the model is applied $\Delta t_{eval}/\Delta t_{train}$ times ($\Delta t_{train}$ is the fast time step between $SSM^{fast}$'s observations).

# E. Additional performance comparisons between TAWM and baseline models

In this section, we provide additional performance comparisons between TAWM and baseline models across varying inference $\Delta t$ values, as shown in Figure 11. On each task, both TAWM variants (using RK4 and Euler integration methods) were trained for 1.5 million steps, with $\Delta t$ values sampled from a Log-Uniform(1 ms, 50 ms) distribution during training. In contrast, the baseline models were trained solely on a fixed default time step of $\Delta t = 2.5$ ms. For these baselines, we use the pretrained weights provided by the original TD-MPC2 paper (Hansen et al., 2024) for each corresponding task.

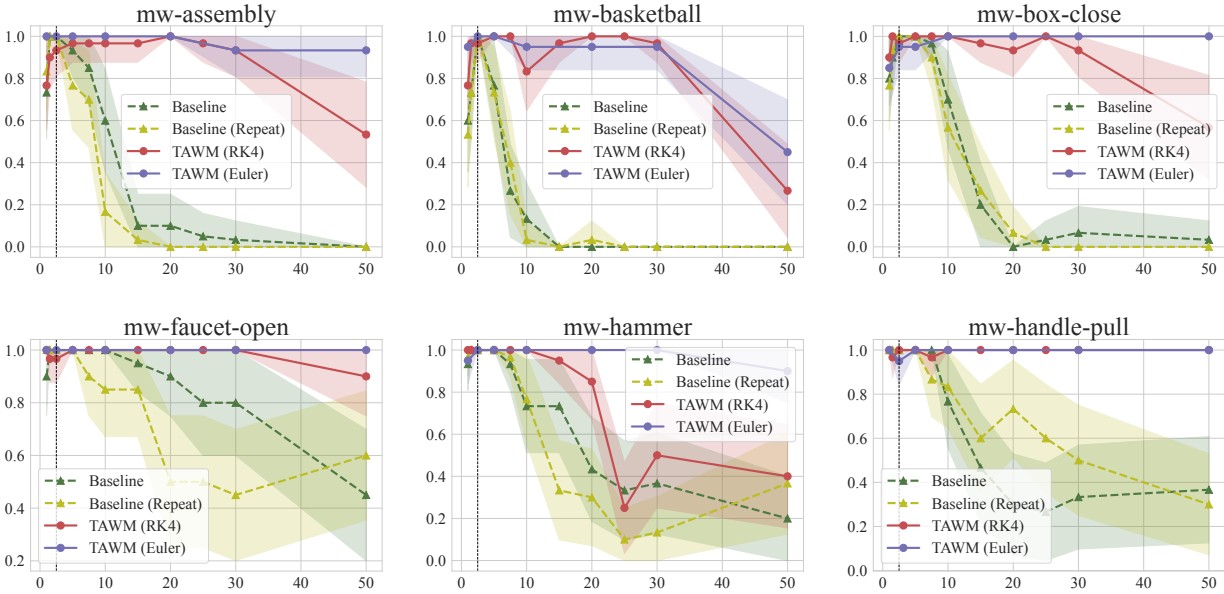

*Figure 11.* **Performance comparisons for 6 additional tasks from Meta-World tasks**. The x-axis corresponds to the evaluation $\Delta t$ (in $ms$), and the y-axis corresponds to success rate. We trained our TAWM model using either RK4 or Euler integration method with log-uniform sampling strategy. The baseline method was trained using only the default time step ($\Delta t = 2.5ms$), which is signified with the black dashed lines. For fair comparisons, we extended the baseline method by repeating the same actions for the larger evaluation time steps than the default one. For instance, when the evaluation $\Delta t = 5ms$, we repeat the same action for $5/2.5 = 2$ times. Plots show mean and 95% confidence intervals over 3 seeds, with 10 evaluation episodes per seed.

# F. Additional learning curve comparisons between TAWM and baseline models

In this section, we present additional learning curve results for TAWM and baseline models on various Meta-World control tasks. Specifically, Figure 12 and Figure 13 show the learning curves of our time-aware models (TAWMs) and the non-time-aware baseline models (original TD-MPC2), evaluated at different inference $\Delta t$ values. Each curve is generated by evaluating the models at various training steps using a fixed inference $\Delta t \in [1, 50]$ ms.

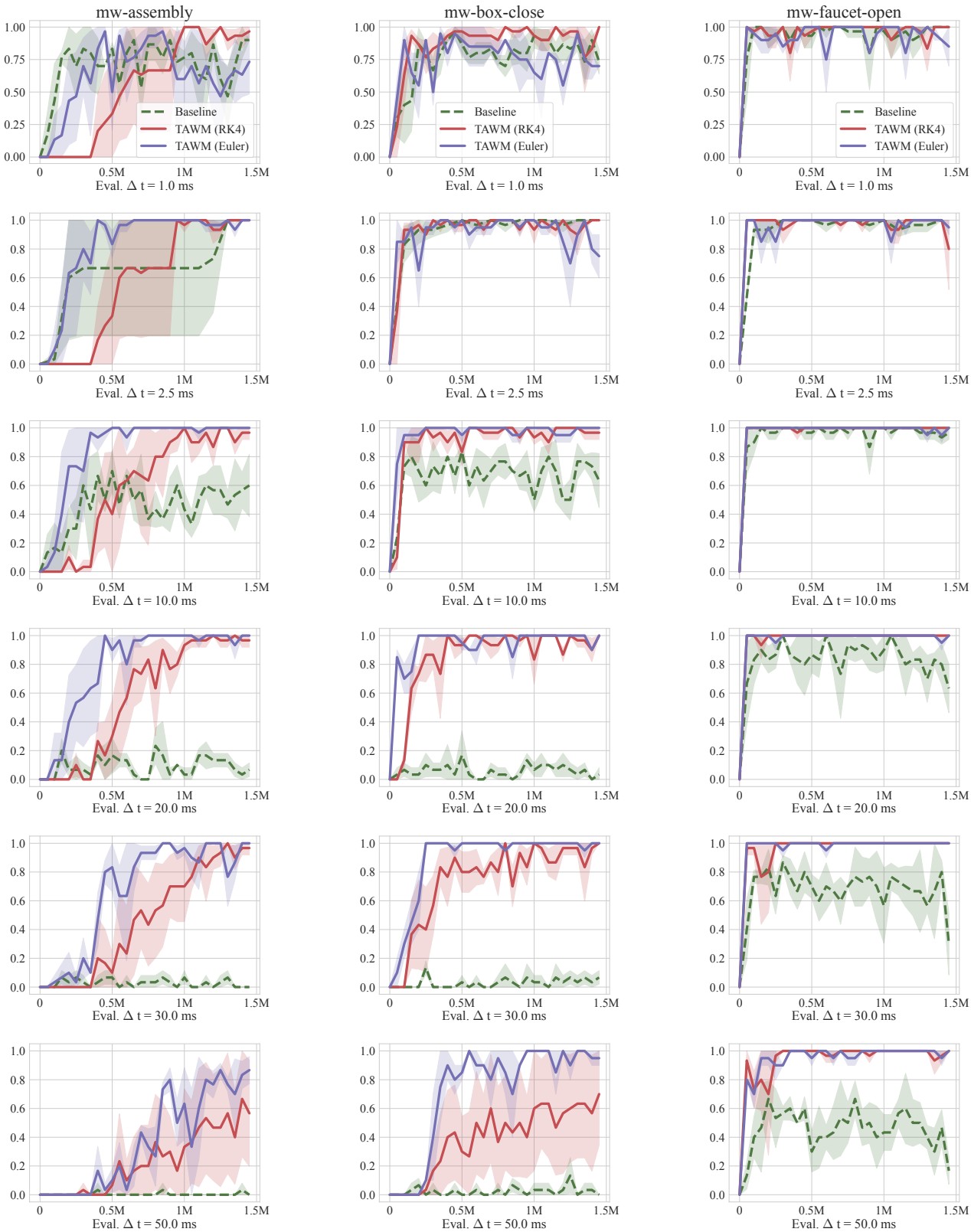

*Figure 12.* **Additional Meta-World tasks: Success Rate Curve under different evaluation time step sizes.**

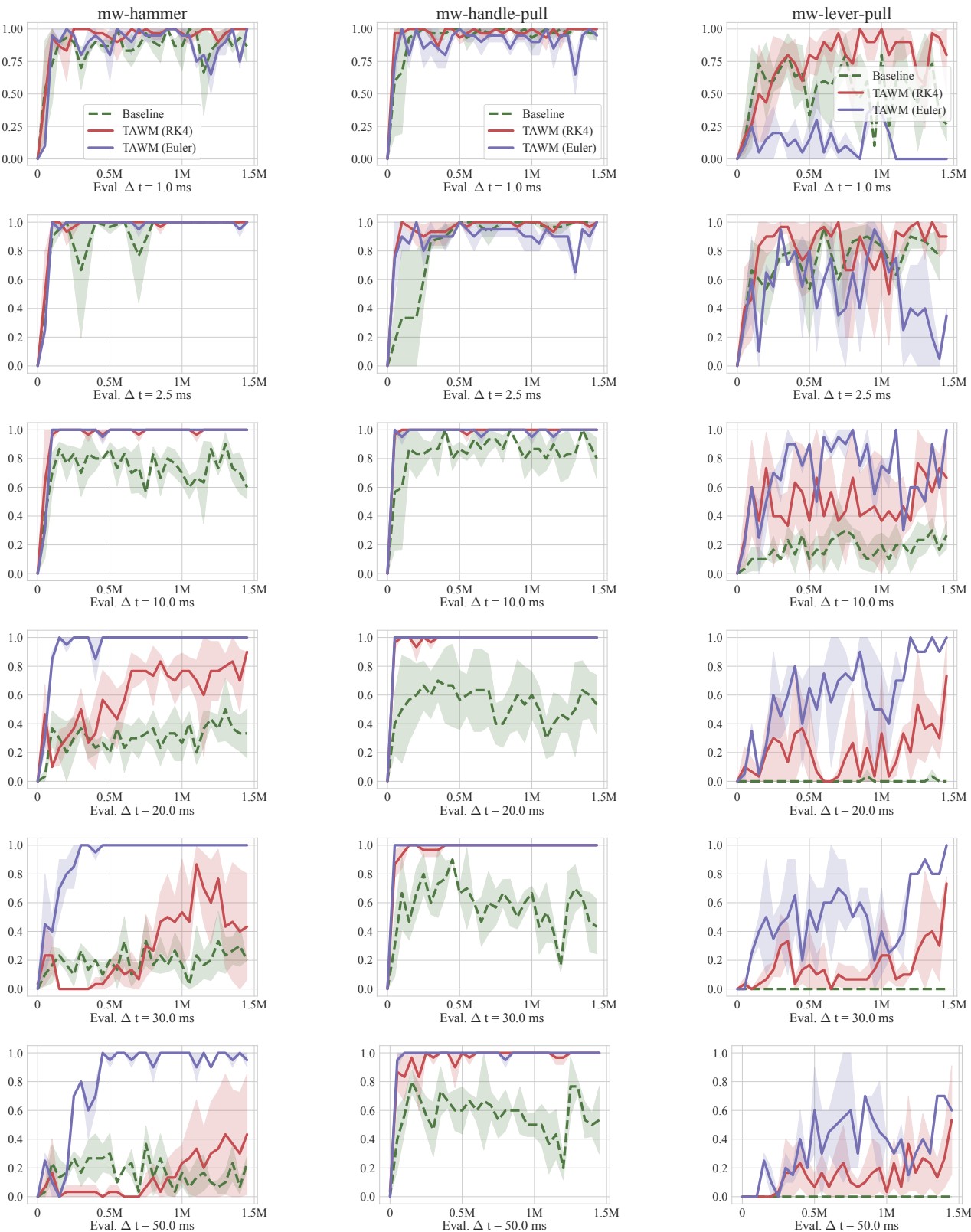

*Figure 13.* **Additional Meta-World tasks: Success Rate Curve under different evaluation time step sizes.**

# G. Additional experiments on PDE control problems.

In this section, we provide detailed descriptions of the PDE control tasks and present additional experiments and analysis of our TAWM on various PDE control problems in `control-gym` environments (Zhang et al., 2024).

## G.1. PDE Control Problems.

The PDE problems are one-dimensional PDE control problems featuring periodic boundary conditions and spatially distributed control inputs. The spatial domain is defined as $\Omega = [0, L] \subset \mathbb{R}$. The continuous field of the PDE is defined as $u(x,t) : \Omega \times \mathbb{R}^+ \to \mathbb{R}$, where $x, t$ represent the spatial coordinates in the field and time, respectively. Generally, the PDE in each control task is defined as:

$$\frac{\partial u}{\partial t} - \mathcal{F}\left(\frac{\partial u}{\partial x}, \frac{\partial^2 u}{\partial x^2}, ...\right) = a(x,t)$$

In the equation above, $\mathcal{F}$ is a differential operator (linear or non-linear) defined differently for each control task. The specific formulations of $\mathcal{F}$ are carefully described in the next subsection for each environment. $a(x,t)$ is the distributed control force over the PDE field. $a(x,t)$ is defined as:

$$a(x,t) = \sum_{j=0}^{n_a - 1} \Phi_j(x)a_j(t)$$

The control force is composed of $n_a$ scalar control inputs $a_j(t)$, each influencing a specific subset of the domain $\Omega$ through its corresponding forcing support function $\Phi_j(x)$. Generally, each action introduces external forces/energy to the PDE fields to control their dynamics and steer them toward the target state. More comprehensive details of the PDE environments are described in (Zhang et al., 2024). In our experiments, the target state of all PDE control tasks is $s_{target} = \vec{0}$. At each step $t$, the reward is computed as the LQ-error between the current state $s_t$ and the target state $s_{target}$:

$$r_t = J(a_t) = -\mathbb{E}\left[(s_t - s_{target})^\top Q(s_t - s_{target}) + a_t^\top R a_t\right]$$

## G.2. Burgers' Equation.

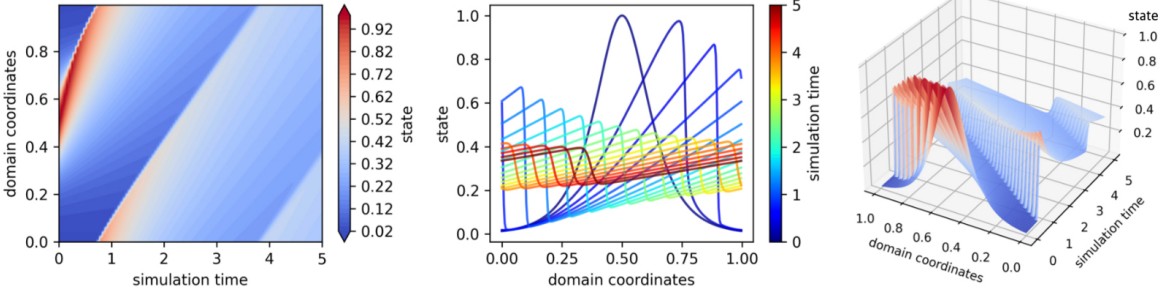

*Figure 14.* Visualizations of systems dynamics of uncontrolled Burgers PDE. The spatial domain has length $L = 1$ with the diffusivity (viscosity) parameter $\nu = 10^{-3}$. The initial state is $u(x, t = 0) = sech(10x - 5)$.

Burgers' equation is a simplified PDE that captures the essential dynamics of fluid waves and gas dynamics. The visualization of the Burgers' Equation is shown in Figure 14. The field velocity $u(x,t)$ in Burgers' equation is defined as:

$$\frac{\partial u}{\partial t} + u\frac{\partial u}{\partial x} - \nu\frac{\partial^2 u}{\partial x^2} = a(x,t)$$

where $\nu > 0$ is the diffusivity (viscosity) parameter, and $a(x,t)$ is a source term modeling an external force acting on the PDE system at coordinate $x$ and time $t$.

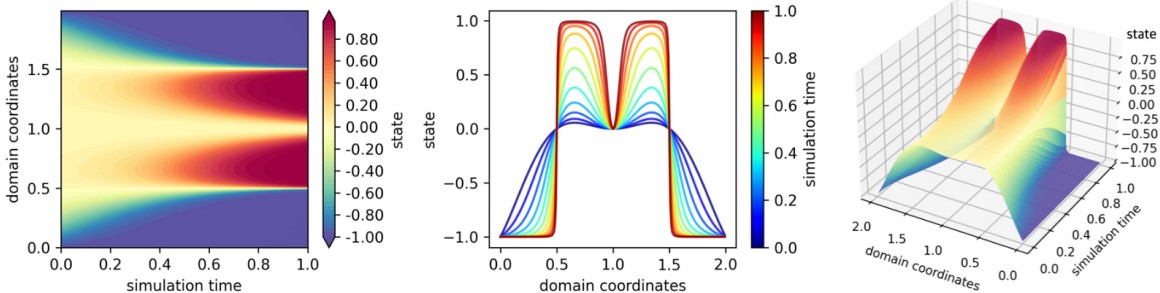

*Figure 15.* Visualizations of systems dynamics of uncontrolled Allen-Cahn PDE. The spatial domain has length $L = 2$ with the diffusivity (viscosity) parameter $\nu = 10^{-4}$ and potential constant $V = 5.0$. The initial state is $u(x, t = 0) = (x - 1)^2 \cdot cos(\pi(x - 1))$.

### G.3. Allen-Cahn Equation.

In material sciences, the Allen-Cahn PDE is a non-linear PDE used to model the binary alloy systems' phase separation. The visualization of the Allen-Cahn equation is shown in Figure 15. The temporal dynamics of field $u(x, t)$ is:

$$\frac{\partial u}{\partial t} - \nu^2 \frac{\partial^2 u}{\partial x^2} + V \cdot (u^3 - u) = a(x, t)$$

where $u = \pm 1$ indicates the presence of each phase, $\nu > 0$ is the diffusivity/viscosity parameter, $V$ is the potential constant, and $a(x, t)$ is a source term modeling an external force acting on the PDE system at coordinate $x$ and time $t$.

### G.4. Wave Equation.

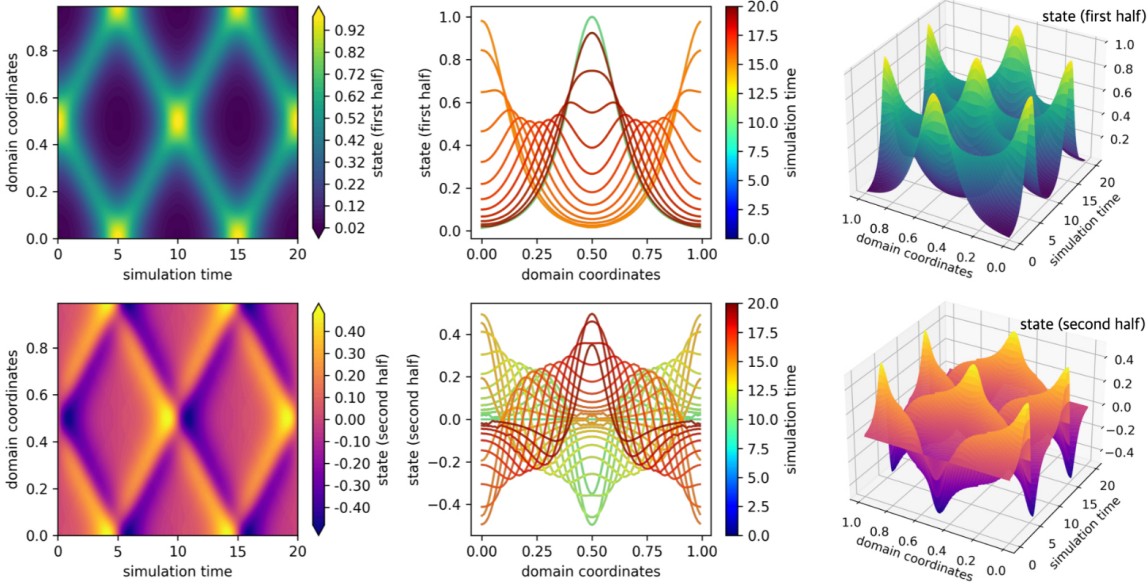

*Figure 16.* Visualizations of systems dynamics of uncontrolled Wave PDE. The spatial domain has length $L = 1$ with $c = 0.1$. The initial state is $u(x, t = 0) = sech(10x - 5)$ and $\psi(x, t = 0) = 0$.

The wave equation is a second-order linear PDE describing the spatial propagation of waves in homogeneous mediums. Such wave propagation PDE has many important applications in physics and engineering problems. The visualization of Wave PDE is shown in Figure 16. The scalar quantity $u(x, t)$ has wave-propagation dynamics defined as:

$$\frac{\partial^2 u}{\partial t^2} - c^2 \frac{\partial^2 u}{\partial x^2} = a(x, t)$$

where $c$ is spatial wave speed, and $a(x, t)$ is a source term of a force acting on the system at coordinate $x$ and time $t$.

## G.5. Additional learning curve comparisons on PDE control tasks.

We present additional learning curves on PDE control tasks to provide insights into TAWM training efficiency on these tasks. Due to the small simulation time step required for stable simulation in PDE-Burgers ($\Delta t_{sim} = 10^{-3}$ seconds), training on this task takes longer in wall-clock time. Therefore, we trained TAWM for only 750k steps on PDE-Burgers. The learning curves of TAWM and the baseline under different observation $\Delta t$ values are shown in Figure 17. For each model and environment, the same set of intermediate weights is used for evaluation at each training step.

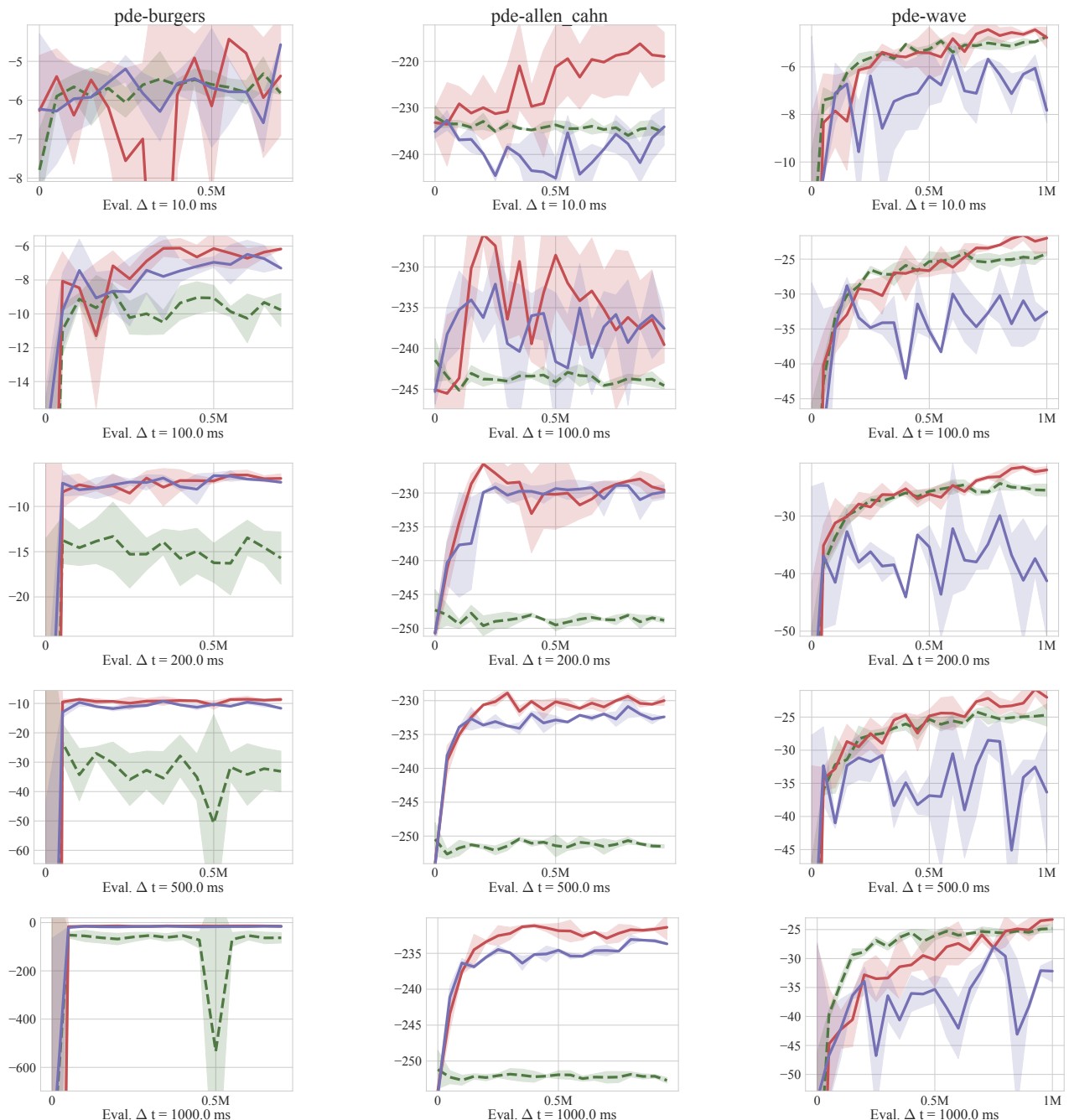

*Figure 17.* **PDE control tasks: Average Negative LQ Error learning curves under different evaluation time step sizes.** The default time step size are $\Delta t = 50\,ms/10\,ms/100\,ms$ for PDE-Burgers, PDE-Allen-Cahn, PDE-Wave, respectively.

# H. Proofs

In this section, we present the proofs of Lemma 4.1 and Lemma 4.2 from the theoretical analysis of TAWM's sample efficiency in Section 4.3.

## H.1. Proof of Lemma 4.1

*Proof.* According to the definitions of the true dynamics function $\bar{f}$ and the optimal learned dynamics function $d^*$, we have

$$z_{t+\Delta t} - z_t = \bar{f}(z_t, a_t, \Delta t)\,\Delta t = d^*(z_t, a_t, \Delta t)\,\tau(\Delta t)$$

$$\Rightarrow \bar{f}(z_t, a_t, \Delta t) = d^*(z_t, a_t, \Delta t)\,\frac{\tau(\Delta t)}{\Delta t}\,. \tag{5}$$

Since we are considering scenarios where the environment dynamics can be captured with $\Delta\bar{t}$, the following holds:

$$||d^*(z_t, a_t, \Delta t) \cdot \frac{\tau(\Delta t)}{\Delta t} - d^*(z_t, a_t, \Delta\bar{t}) \cdot \frac{\tau(\Delta\bar{t})}{\Delta\bar{t}}|| < \epsilon$$

$$\Rightarrow ||d^*(z_t, a_t, \Delta t) - d^*(z_t, a_t, \Delta\bar{t}) \cdot \frac{\Delta t}{\tau(\Delta t)} \cdot \frac{\tau(\Delta\bar{t})}{\Delta\bar{t}}|| < \epsilon \cdot \frac{\Delta t}{\tau(\Delta t)}. \tag{6}$$

Given the sampling-frequency assumption in Section 4.3 ($\Delta t \in [0.001, 1.0]$) and the definition of $\tau(\cdot)$, we observe that $\frac{\Delta t}{\tau(\Delta t)} < 1$. Consequently, Lemma 4.1 holds.

$\square$

## H.2. Proof of Lemma 4.2

*Proof.* Under Assumption 3 in Section 4.3, the following holds for our current dynamics model $d$:

$$||d(z_t, a_t, \Delta t) - d(z_t, a_t, \Delta\bar{t}) \cdot \frac{\Delta t}{\tau(\Delta t)} \cdot \frac{\tau(\Delta\bar{t})}{\Delta\bar{t}}|| < \epsilon. \tag{7}$$

If we denote $I(\Delta t) = \frac{\Delta t}{\tau(\Delta t)} \cdot \frac{\tau(\Delta\bar{t})}{\Delta\bar{t}}$, then the following holds:

$$
\begin{aligned}
&||d(z_t, a_t, \Delta t) - d^*(z_t, a_t, \Delta t)|| \\
=\ &||d(z_t, a_t, \Delta t) - d(z_t, a_t, \Delta\bar{t}) \cdot I(\Delta t) + d(z_t, a_t, \Delta\bar{t}) \cdot I(\Delta t) \\
&- d^*(z_t, a_t, \Delta t) + d^*(z_t, a_t, \Delta\bar{t}) \cdot I(\Delta t) - d^*(z_t, a_t, \Delta\bar{t}) \cdot I(\Delta t)|| \\
\leq\ &||d(z_t, a_t, \Delta t) - d(z_t, a_t, \Delta\bar{t}) \cdot I(\Delta t)|| \\
&+ ||d^*(z_t, a_t, \Delta t) - d^*(z_t, a_t, \Delta\bar{t}) \cdot I(\Delta t)|| \\
&+ ||d(z_t, a_t, \Delta\bar{t}) - d^*(z_t, a_t, \Delta\bar{t})|| \cdot I(\Delta t) \\
=\ &2\epsilon + ||d(z_t, a_t, \Delta\bar{t}) - d^*(z_t, a_t, \Delta\bar{t})|| \cdot I(\Delta t).
\end{aligned}
\tag{8}
$$

Likewise, the modeling error at time step $\Delta t$ is upper bounded by the error at $\Delta\bar{t}$. Therefore, by minimizing the error at $\Delta\bar{t}$, we expect that the error at the smaller time step would safely decrease as well. $\square$

