# OpenReview forum: "Time-Aware World Model for Adaptive Prediction and Control"
_ICML.cc/2025/Conference — ICML 2025 poster_

### Official Review · Reviewer_ojxn · 2025-03-02

**Overall Recommendation:** 4

**Summary:**

The work presents a world model conditioned on time step size, showing that training with a sampling of different time step sizes can improve long horizon prediction stability. The algorithm conditions the world model on the time step size and uses 4th order Runge-Kutta to integrate the dynamical model. Experiments demonstrate performance improvements across a variety of simulated robotics tasks when using the world model for model-predictive control with different observation rates without increasing training budget (in data or steps). The new model out-performs an existing world model that models two different time scales.

## update after rebuttal

Increased to accept based on the greater statistical rigor with some promising results, combined with clear text to address my comments.

**Claims And Evidence:**

The claims and their evidence:
- claim: Training on variable time step sizes (TAWM) addresses compounding errors with one-step prediction.
	- evidence: The MetaWorld experiments examine error when sampling inference with long time steps, showing success when using long time steps.
- claim: TAWM handles variable frame rate data without altering training budget.
	- evidence: The MetaWorld experiments include many comparisons of the TAWM to fixed time step models on varying scales. In most cases TAWM does better, primarily when facing longer time steps (lower frequency). The main exception is the hammer environment.
- claim: TAWM works in control tasks with different observation rates without increasing training budget (data or steps)
	- evidence: Same evidence as above.
- claim: (implicit) Euler method fails where 4th order Runge-Kutta succeeds
	- evidence: None provided. This merits a separate evaluation or ablation (perhaps in the appendix).

There is an implied claim that log-uniform sampling is important for the training process, but the evidence shows some advantages to uniform time step sampling at long horizons, with mixed evidence of differences at short horizons (overlapping confidence intervals). See Figures 7 and 8.

**Essential References Not Discussed:**

It would be reasonable to reference AlphaGo and subsequent work (AlphaGo Zero, AlphaZero, MuZero, Muesli, and so on) in that model family as exemplars of the model-based RL approach. The latter models in this series developed model-based RL algorithms similar to the Dreamer model family with very strong performance across multiple domains.

DayDreamer (Wu et al. in CoRL 2023) may merit consideration as a robotics model that trains in simulation alongside a physical robot (https://proceedings.mlr.press/v205/wu23c.html). This handles the online learning case, which contrasts with the simulation work in this paper.

**Experimental Designs Or Analyses:**

Yes. The Meta-World world evaluations.
- These only have an ablated version of the TAWM as comparison. They would benefit from alternative methods.
- No experiments report on inference time requirements.
- Only one baseline model was compared.

**Methods And Evaluation Criteria:**

Yes. Robotics tasks are a natural scenario for long-range control tasks (using MPC or otherwise). Testing across different time horizons is the natural task to test.

**Other Comments Or Suggestions:**

"To assess our time-aware model’s performance at different inference-time observation rates, we evaluated it on multiple tasks with varying ∆ t. As shown in Figure 12, it outperforms the baseline (trained at a fixed ∆ t = 2.5ms) in across three tasks"
- This should be Figure 2 (in the body).

**Other Strengths And Weaknesses:**

# strengths
- originality: The core idea of conditioning on variable time steps combines simplicity with novelty, showing how to generally improve world modeling tasks.
- significance: Improving world model temporal behavior is valuable across a number of task domains in robotics and other physical systems. Enabling models to handle variable time steps is also relevant to other architectures like directly learning behavioral policies (instead of planning over a time step aware model).

# weaknesses
- clarity: A few points were not clearly addressed, including: the benefit from using 4th order Runge-Kutta instead of Euler integration and the lack of substantial differences between log-uniform and uniform sampling.
	- The Nyquist sampling motivation is also weak given the lack of a theoretical or empirical method that establishes a connection to the theoretical signal optimization needs.

**Questions For Authors:**

- [Q1] How much of the performance is due to the use of RK4 instead of Euler's method?
- [Q2] What evidence is there that log-uniform sampling is superior to uniform scaling?
	- Figures 7 and 8 show uniform sampling works better at long time steps. Short time steps overlap in performance with the log-uniform case as well. The claims ultimately state that any sampling can be used. If this is the claim being made that should be explicit at the start of the paper.
- [Q3] How fast is inference / planning in TAWM?
	- Considering the downstream task is robotic manipulation having a sense of how fast model predictions and planning occur is important to assess deployment in the real world (eventually).

**Relation To Broader Scientific Literature:**

The work is contextualized in relation to the model-based reinforcement learning literature. The proposed model performs model-predictive control in a world model, rather than learning a control policy (as in the Dreamer models).

The key contribution is devising a world model training approach that can train on variable time step durations and perform inference when observations occur at different frame rates. This is contrasted with the existing MTS3 model that is trained for a discrete number of temporal resolutions (two in the existing model).

**Theoretical Claims:**

No. No proofs were made.

---

> ### Author Rebuttal · Authors · 2025-03-30
>
> We sincerely appreciate your comprehensive reviews and suggestions!
>
> **Weakness 1.** The Nyquist sampling motivation is weak given the lack of a theoretical or empirical method to establish the connection to the theoretical signal optimization needs.
>
> A1. We have added the theoretical analysis of the sample efficiency and effectiveness of our proposed time-aware world model. Due to the word limit, we would like to refer to **our answer A1 to reviewer AknR** above for the theorem, lemmas, and their high-level idea for theoretical analysis. We would be happy to provide the proof details if the reviewer requests it in the next rebuttal.
>
> ----
> **Weakness 2.** A few points were not clearly addressed, including the benefit of using 4th-order Runge-Kutta instead of Euler and the lack of substantial differences between log-uniform and uniform sampling.
>
> A2. Since this concern overlaps with the reviewer's **Q1** and **Q3**, we would like to refer to our answers A5 and A6 in this rebuttal.
>
> ----
> **Suggestion 1.** Additional references.
>
> A3. We agree that AlphaGo and its subsequent works as well as DayDreamer are indeed relevant to this paper. We have included these works in our references and will be available in the final version.
>
> ----
> **Suggestion 2.**  This (Fig. 12) should be Fig. 2 (in the body).
>
> A3. We have updated the main text accordingly to refer to both Figure 2 and Figure 12.
>
> ----
> **Q1.** How much of the performance is due to the use of RK4 instead of Euler's method?
>
> A6. Thank you for raising your concern regarding the choice integration method. We have included additional ablation studies on the RK4 vs Euler integration method in our anonymous website: https://sites.google.com/view/anonymous-site-rebuttal-6714. We also include additional experiments on `PDE-Control` tasks from the `control-gym` envs: arxiv.org/abs/2311.18736.
>
> We emphasize that our proposed method’s central contribution is the adaptive time stepping, which is **integration-agnostic** and can be used with any integration, depending on the nature of task dynamics.
>
> We employed the RK4 integration method because it is generalizable to both systems with simple, linear dynamics and highly complex, nonlinear dynamics and is a standard method for simulations of physical systems. For most robot manipulation tasks (Meta-World), the dynamics are sufficiently simple to be approximated with the Euler integration method. Empirically, our ablation shows that TAWM Euler performed better than RK4-based TAWM in most simple Meta-World tasks, suggesting the underlying dynamics of Meta-World tasks are sufficiently simple to be captured by the Euler integration method. The advantages of RK4 are more apparent in tasks with complex, non-linear dynamics, such as the PDE-Control tasks.
>
> These results reinforce the merit of our core contribution of adaptive time stepping for training world models. The integration method and $\Delta t$-sampling method are two parameters we can adjust to maximize TAWM's performance and efficiency --both of which outperform the baseline.
>
> ----
> **Q2.** What evidence is that log-uniform sampling is superior to uniform scaling? The claims ultimately state that any sampling can be used. If this is the claim being made that should be explicit at the start of the paper.
>
> A7. Thank you for your comments! As explained in A6, the integration method and $\Delta t$-sampling method are two parameters that we can adjust to maximize TAWM's performance and efficiency. Therefore, uniform sampling and/or the Euler method can be better choices depending on the tasks. For example, uniform sampling performs better than log-uniform sampling in tasks with sufficiently slow dynamics to be captured by $\Delta t_{max}$. For tasks demanding fast inference time with sufficiently simple dynamics, Euler method is preferred.
>
> We will be sure to make this point much more clear at the beginning of our paper in the final revision.  Thank you for the suggesiton.
>
> ----
> **Q3.** How fast is inference/planning in TAWM?
>
> A8. We provide additional details of the inference time below. Inference time is averaged over 1000 planning steps for each model.
> * Baseline: $\mu=$0.027 s; Q1 = 0.026 s; Q3 = 0.027 s
> * Euler: $\mu=$0.028 s; Q1 = 0.028 s; Q3 = 0.028 s
> * RK4: $\mu=$0.048 s; Q1 = 0.048 s;  Q3 = 0.05 s
>
> The tradeoff between Euler and RK4 is well-known in simulation: Euler is faster but can be unstable/less accurate than RK4 depending on the underlying dynamics. As mentioned in our answers A6 and A7, the integration method is one of the adjustable parameters. If underlying dynamics are sufficiently simple and slow to be approximated by the Euler method, the Euler method is preferred. Otherwise, RK4 is a more generalizable method.
>
> We have incorporated your comments and new results into our paper. We thank you again for your comprehensive reviews and valuable suggestions, and we hope our answers sufficiently address your concerns and questions.

---

> > ### Comment · Reviewer_ojxn · 2025-04-02
> >
> > Thank you for comprehensively addressing my comments. Only adding remarks about open topics below.
> >
> > # Q1
> >
> > Are there any statistical tests of differences to report for the results linked in the website? From visual inspection the `PDE-Control` problems look quite close between Euler & RK4. I agree that integrator choice is a hyperparameter to tune. It would still be helpful to quantify the difference (or lack thereof) in the results.
> >
> > # Q2
> >
> > As with Q1, it would help to have statistical analyses supporting these claims.
> >
> > # Suggestion 1.
> >
> > Can you provide the new text on these related works in the next rebuttal?

---

> > > ### Author Response · Authors · 2025-04-02
> > >
> > > Thank you for your timely questions! We would like to address your questions below:
> > >
> > > ----
> > > **Q1.Statistical Test of Euler vs RK4 integration**
> > >
> > > We used one-side pairwise t-tests to compare the rewards of Euler and RK4 TAWM on PDE-Control tasks.. We consider p-value < 0.01 to be significant. T-value > 0 indicates Euler performs better than RK4 and vice versa.
> > >
> > > **pde-allen_cahn:**
> > > * dt = 0.01: stats = -9.48; p-value = 0.0	[SIGNIFICANT]
> > > * dt = 0.05: stats = -1.62; p-value = 0.11
> > > * dt = 0.1: stats = 2.44; p-value = 0.02
> > > * dt = 0.5: stats = -0.16; p-value = 0.88
> > > * dt = 1.0: stats = -0.73; p-value = 0.47
> > >
> > > **pde-burgers:**
> > > * dt = 0.01: stats = 0.02; p-value = 0.99
> > > * dt = 0.05: stats = 1.29; p-value = 0.2
> > > * dt = 0.1: stats = -0.01; p-value = 0.95
> > > * dt = 0.5: stats = -2.85; p-value = 0.01	[SIGNIFICANT]
> > > * dt = 1.0: stats = -5.23; p-value = 0.0	[SIGNIFICANT]
> > >
> > > **pde-wave:**
> > > * dt = 0.01: stats = 0.33; p-value = 0.74
> > > * dt = 0.05: stats = 0.94; p-value = 0.35
> > > * dt = 0.1: stats = -1.7; p-value = 0.1
> > > * dt = 0.5: stats = -0.1; p-value = 0.92
> > > * dt = 1.0: stats = -3.24; p-value = 0.0	[SIGNIFICANT]
> > >
> > > For PDE-Control tasks, all the significant t-tests indicate that Euler underperforms (t-value < 0) compared to RK4 integration. On the other hand, for tests with t-value > 0, the p-value is not sufficiently significant to confirm Euler performs better than RK4 in PDE-Control tasks. These tests indicate overall, for PDE-Control tasks, RK4 is likely a better integration method.
> > >
> > > ----
> > > **Q2.Statistical Test of Uniform vs Log-Uniform Sampling**
> > > Due to the word limits, we focus on 4 tasks, all of which we have included in our ablation study in the paper. Since the evaluation metric of MetaWorld tasks is `success_rate`, we used the one-side Fisher exact tests to test the statistical significance between the performance of TAWMs trained with Uniform and Log-Uniform sampling. The alternative hypothesis is Uniform > Log-Uniform for Meta-World tasks.
> > >
> > > **mw-assembly:**
> > > * dt=0.001: stats = 0.64; p-value = 0.86
> > > * dt=0.0025: stats = 0.23; p-value = 0.99
> > > * dt=0.01: stats = 1.0; p-value = 0.69
> > > * dt=0.02: stats = 0.48; p-value = 0.88
> > > * dt=0.03: stats = 1.55; p-value = 0.5
> > > * dt=0.05: stats = 2.65; p-value = 0.06
> > >
> > > **mw-basketball:**
> > > * dt=0.001: stats = 0.49; p-value = 0.95
> > > * dt=0.0025: stats = 2.07; p-value = 0.5
> > > * dt=0.01: stats = 1.62; p-value = 0.37
> > > * dt=0.02: stats = 1.0; p-value = 0.75
> > > * dt=0.03: stats = 1.0; p-value = 0.69
> > > * dt=0.05: stats = 13.8; p-value = 0.0	[SIGNIFICANT]
> > >
> > > **mw-hammer:**
> > > * dt=0.001: stats = 0.21; p-value = 0.98
> > > * dt=0.0025: stats = 1.48; p-value = 0.65
> > > * dt=0.01: stats = 1.48; p-value = 0.65
> > > * dt=0.02: stats = 6.89; p-value = 0.08
> > > * dt=0.03: stats = 31.0; p-value = 0.0	[SIGNIFICANT]
> > > * dt=0.05: stats = 45.31; p-value = 0.0	[SIGNIFICANT]
> > >
> > > **mw-lever-pull:**
> > > * dt=0.001: stats = 0.68; p-value = 0.83
> > > * dt=0.0025: stats = 0.97; p-value = 0.69
> > > * dt=0.01: stats = 1.09; p-value = 0.55
> > > * dt=0.02: stats = 4.33; p-value = 0.01	[SIGNIFICANT]
> > > * dt=0.03: stats = 21.0; p-value = 0.0	[SIGNIFICANT]
> > > * dt=0.05: stats = 1.3; p-value = 0.4
> > >
> > > The statistical tests indicated that there is a lack of differences between the two sampling methods on `mw-assembly`, while Uniform sampling is generally better on other tasks, especially at larger $\Delta t$.
> > >
> > > ----
> > > **Suggestion 1**
> > > **At the beginning of Introduction:** Deep reinforcement learning (DRL) has recently demonstrated expert-level or even superhuman capabilities on many highly complex and challenging problems, such as `\cite{AlphaGo}` and `\cite{AlphaGo_Zero}`  in Go games, `\cite{AlphaZero}` in Chess and Shogi games,  `\cite{MuZero}` in multiple games (Atari, Chess, Go), and `\cite{AlphaStar}` for StarCraft II. Beyond games, DRL has also shown ground-breaking performance in scientific discoveries such as protein prediction `\citep{AlphaFold, AlphaProteo}`, solving IMO-level geometries `\citep{AlphaGeometry}`, and sorting algorithms `\citep{AlphaDev}`.
> > >
> > > **In Related Works, just before MTS3:** Dreamer models have demonstrated their capability in learning and deploying directly on physical robots `\citep{DayDreamer}`, showing the potential of world models in physical control tasks. While DayDreamer can train models for manipulation tasks at low sampling frequencies, their robot motions between time steps are very slow, as shown in their demo. Such slow motions, while stabilizing training, result in slow robot action in practice.
> > >
> > > ----
> > > In addition to updated text on related work, as requested, we will also include the technical explanation of the choices of integration scheme and time steps from the rebuttal in the final revision and additional supplementary materials.
> > >
> > > We sincerely thank you again for your valuable comments and feedback that allowed us to clarify important technical contributions of this work! If you have a chance to review our rebuttal and believe we have effectively addressed your remaining questions and concerns, we would be grateful if you could consider updating your score.

---

### Official Review · Reviewer_V9z9 · 2025-03-11

**Overall Recommendation:** 3

**Summary:**

This paper proposes Time-Aware World Models (TAWM) to enhance the robustness of world models in various frequency control tasks. During training, TAWM takes the randomly sampled transition time interval ($\Delta t$) as an additional condition, enabling it to adapt to the control frequency of the test environment during evaluation. The authors validate the effectiveness and robustness of TAWM through experiments on the MetaWorld benchmark.

**Claims And Evidence:**

I am confused about the motivation of this paper. The authors claim that learning a time-aware model is essential to address issues like temporal resolution overfitting and inaccurate system dynamics caused by differences in observation frequencies during training and testing. However, the manipulation tasks in the experiments do not seem to encounter these problems. Therefore, the authors should provide more examples of tasks that face these challenges and validate their approach on a wider range of benchmarks.

**Essential References Not Discussed:**

No.

**Experimental Designs Or Analyses:**

Yes.

**Methods And Evaluation Criteria:**

Yes.

**Other Comments Or Suggestions:**

No.

**Other Strengths And Weaknesses:**

***Strengths***
1. This paper is clearly written and easy to follow.
2. The proposed method is reasonable, and the experimental results demonstrate TAWM's robustness to control frequency and the efficiency of policy learning.

***Weaknesses***
1. This method lacks novelty, as the model learning in TAWM is largely based on TDMPC-2, with the main difference being the addition of time interval as an input.
2. There is a gap between the experimental design and the motivation of the method. In most manipulation tasks, the data collection frequency matches the testing frequency. The authors need to provide a clearer explanation of which tasks experience discrepancies between training and testing frequencies and conduct experiments on the relevant benchmarks.
3. The experimental comparison is unfair, as the baseline, TDMPC-2, is trained using a single control frequency, while TAWM is trained with data from various frequencies.

**Questions For Authors:**

See weaknesses.

**Relation To Broader Scientific Literature:**

This paper presents an efficient, model-agnostic approach to training Time-Aware World Models (TAWM) that adapts to varying control frequencies without increasing sample complexity, reducing the need for retraining and lowering computational costs.

**Theoretical Claims:**

Yes.

---

> ### Author Rebuttal · Authors · 2025-03-30
>
> Please see additional experimental results [here](https://sites.google.com/view/anonymous-site-rebuttal-6714).
>
> **Q1. Gap between experimental design and motivation (data collection frequency matches the testing frequency in most manipulation tasks).**
>
> A1. We appreciate the reviewer’s concerns but believe this is a misunderstanding of our work. While data collection frequency can match testing frequency in laboratory settings, **this is not always feasible or effective in practice**.
>
> Prior works have acknowledged such limitations [1]. Consider training a model at $\Delta t=$ 2.5 ms and deploying it on a physical robot with sensors operate at 100 fps ($\Delta t=$10 ms). One could either (1) use the trained model directly or (2) repeat the same action for 4 substeps to compensate for the mismatch. However, as Fig. 2 shows, TAWM outperforms these strategies across various frequencies, making it flexible for any $\Delta t$ encountered in practice w/o additional data or training.
>
> One might argue that we could train the model at 10 ms in simulation to match the real-world frequency.  Figure 4 shows that this solution is not robust when $\Delta t$ grows larger, as larger $\Delta t$ introduces instability due to missing important high-frequency dynamics (see Introduction).  TAWM overcomes this issue by concurrently sampling multiple frequencies with better performance.
>
> While approaches like DayDreamer [2] can train models for manipulation tasks at low sampling frequencies, their robot motions between time steps are very slow, as shown in their demo. Such slow motions prevent the loss of important dynamics and stabilize training, but result in slow robot actions in practice. In contrast, our TAWM can effectively learn to solve tasks at large $\Delta t$ without limiting the robot's motion speed, effectively learning both fast and slow task dynamics simultaneously.
>
> We further conducted experiments on PDE-control environments, whose dynamics fundamentally differ from manipulation tasks (see **A6 of Reviewer 3**). Our results indicate that TAWM is effective beyond manipulation tasks and generalizes well across different classes of control problems.
>
> ----
> **Q2. Lack of novelty and unfair comparison**
>
> A2. We respectfully disagree with the reviewer’s comment that the comparison is unfair. **Our training strategy—sampling multiple frequencies—is precisely the contribution of our work**. The baseline TD-MPC2 model doesn't incorporate the temporal element, so it cannot be trained the same way.  A comparison with TD-MPC2 is either (1) use it as-is or (2) manually apply substeps and repeat the same action (see A1).  TAWM outperforms both baselines by a considerable margin in as fair a comparison as possible, given the fundamental difference in approaches.
>
> Regarding novelty,  one might view our primary modification as simply adding adaptive time intervals to the model input. However, we do not believe this should be dismissed as a lack of novelty for the following reasons:
>
> * **Simplicity with Clear Benefits**:  Often simpler approaches are preferable when they offer clear advantages. Even if the idea may appear 'simple' at first, our extensive experiments demonstrate that it yields substantial performance gains over the baseline without increasing model size or the required training samples. This simplicity also means that other world-model learning methods can readily adopt our approach.
>
> * **Integration of Physical Dynamics and Temporal Elements**: Prior work has generally neglected explicit temporal modeling and adherence to basic physics principles in world models. Simply adding a time interval input, $z_{t+Δt}​=d(z_t​,a_t​,Δt)$, does not ensure compliance with these principles. In contrast, TAWM incorporates two key, model-agnostic components from physics simulation—time stepping and an integration method—that have been overlooked by previous methods. By integrating these components, we efficiently train our dynamics model by reducing the optimization space (e.g., the state remains unchanged when
> $\Delta t = 0$). Our main contribution lies in embedding physical dynamics into world models—a concept applicable across various architectural designs. We emphasize that our method is model-agnostic, and our contribution is in the novel consideration of BOTH *temporal* and *physical dynamics* in the world model -- a significantly new concept that none of the prior works has explored.
>
> We also support our experimental results by theoretical proofs, as requested by the reviewers.
>
> ----
> **Q3. Lack of supplementary material.**
>
> A3. We already provided an appendix with additional experimental results across all environments.  We refer this reviewer to the supplementary material and hope that it addresses any concerns regarding the breadth of our extensive evaluation.
>
> ----
>
> [1] Thodoroff et al., "Benchmarking real-time reinforcement learning."
>
> [2] Wu, Philipp, et al. "Daydreamer: World models for physical robot learning."

---

> > ### Comment · Reviewer_V9z9 · 2025-04-03
> >
> > Thank you for your response! I am curious about how TAWM handles the example provided by the authors—collecting at only 400 FPS and deploying with a 100ms interval. According to Algorithm 1, TAWM requires collecting data at different sampling frequencies during training, and the authors evaluate it on in-distribution frequencies seen in the training set.
> >
> > Moreover, suppose the sampling frequency can be arbitrarily chosen during data collection, and our goal is to obtain an optimal policy. In that case, it seems reasonable to simply fix the collection frequency to match the deployment frequency. From the experimental results, it is evident that in the MetaWorld tasks, when $\Delta t$ is set to 2.5ms (the default value in the simulator), TDMPC2 achieves performance comparable to TAWM on almost all tasks, significantly outperforming lower-frequency policies.

---

> > > ### Author Response · Authors · 2025-04-04
> > >
> > > Thank you for your follow-up questions!
> > >
> > > ----
> > > **Q1.** We'd like to clarify on the example scenario of mismatch between the data collection frequency (400 FPS) and testing frequency (100 FPS).   There appears to be some misunderstanding.  Our TAWM does **not** only collect the data at 400 FPS for training but instead a **mixture of different sampling frequencies**. **If we only trained on 400FPS, it’s essentially the baseline (which is the blue curves in Figs. 2,4,5,6,7,8,12,13,14) and that is not TAWM.**
> > >
> > > Our **core proposal/theoretical motivation is that we can sample data from any arbitrary frequency during the training process**. Using **mixture of sampling frequencies is precisely our main contribution**, which is shown **empirically and theoretically to be more effective and sample-efficient without needing additional data or resources**. We applied this proposed technique to the baseline TD-MPC2 for fair comparisons. Like most existing world models, it does not consider temporal elements as we do in modeling the state transition, leading to potential frequency mismatch in scenarios like the previous example.  Possible strategies for handling this mismatch using the baseline TD-MPC2 would be to use TD-MPC2 sample data in one of these strategies:
> > > 1. Sample at 400 FPS in training, and deploy the model to the test environment (100 FPS).
> > > 2. Sample at 400 FPS in training, and take 4 substeps for one step in the test environment (100 FPS).
> > > 3. Sample at 100 FPS in training, and deploy the model to the test environment (100 FPS).
> > > As we already discussed, the experimental results in Figure 2 and Figure 12 (in the appendix) show that TAWM outperforms strategies (1) and (2) across multiple testing frequencies, including the default frequency used by the baseline method to collect data.
> > >
> > > For strategy (3), please see our answer below.
> > >
> > > ----
> > > **Q2.** This corresponds to strategy (3) above – if we can sample data from an arbitrary frequency, we can sample data from the testing frequency and train the baseline model.
> > >
> > > **First**, we have already shown **this approach is effective only for a sufficiently high frequency, such as 400 FPS ($\Delta t=$ 2.5 ms) in Figure 4**. Specifically, in `mw-assembly`in Figure 4, we trained the baseline models, each with a fixed sampling frequency. The yellow curve (the baseline model trained with only 100 FPS) performs much worse than the green and blue line (baseline models trained with 1000 FPS and 400 FPS, respectively), for the *testing frequency of 100 FPS* (the x-axis corresponds to the time step; 100 FPS is 10 ms on the x-axis). Our Figure 4 shows that **training directly on lower test frequency of 100 FPS and 20 FPS leads to complete failure of 0% success rate.** In contrast, our *TAWM (trained on a mixture of multiple frequencies) in red shows the best performance overall*. We already discussed these results in the subsection **Effects of using Mixtures of Time Step Sizes** on page 6.
> > >
> > > **Second, even if strategy (3) were effective, our TAWM offers superior efficiency**. Why do we have to train multiple models for different testing frequencies when we can train our TAWM **ONCE** and deploy it for different testing frequencies with the same training steps? As demonstrated in Figure 5 (and Figures 13-14 in the appendix), TAWM converges to optimal policies as quickly as the baseline, even at the default test frequency. There's no justification for training multiple specialized models when a single TAWM, with similar computational cost to one baseline model, can adapt to different test frequencies.
> > >
> > > **Third**, as we have included additional results in the anonymous website in the previous response, we want to clarify that the sampling method (Uniform/Log-Uniform/etc.) and integration method (Euler/RK4) are two tunable parameters (please see our responses to reviewer 3’s **Q1,Q2** for more details). As shown in our paper and the additional new results as requested, using TAWM effectively obtain 90-100% success rates across most test frequencies in most Meta-World tasks. This demonstrates the effectiveness of TAWM when the appropriate integration method is properly used.
> > >
> > > **Additionally**, for the theoretical explanation of why training on a mixture of frequencies is more effective and efficient, please see *our response (A1) to Reviewer 1*.
> > >
> > > ----
> > > **SUMMARY:** We emphasize that the key benefit of TAWM is its ability to train ANY dynamical system **just ONCE using a mixture of multiple frequencies** at the same cost to SOTA methods trained at some fixed frequency.  YET, TAWM can be deployed and tested on ANY dynamical system of any testing frequency, ***without training multiple times*** **using multiple testing frequencies**, as in using the SOTA method.
> > >
> > > Thank you for the opportunity to clarify and emphasize our contribution!  If our responses addressed your questions and you've a chance to review our supplementary document. we'd be grateful if you could consider updating your score.

---

### Official Review · Reviewer_AknR · 2025-03-13

**Overall Recommendation:** 3

**Summary:**

This work introduces Time-Aware World Model (TAWM), a model-based approach designed to explicitly incorporate the temporal dynamics of environments. By conditioning on the time step size, ∆t, and training over a diverse range of values ∆t – rather than relying on a fixed time step size – TAWM enables learning of both high- and low-frequency task dynamics in diverse control problems.

**Claims And Evidence:**

The authors claim TAWM efficiently trains the world model M to accurately capture the underlying task dynamics across varying time step size ∆t’s without increasing sample complexity.

The authors demonstrate the results on diverse control problems in MetaWorld environments.

**Essential References Not Discussed:**

Most references have been included in my opinion.

**Experimental Designs Or Analyses:**

Empirically, the authors show that our time-aware world model can effectively solve various control tasks under different observation rates without increasing data nor training steps.

**Methods And Evaluation Criteria:**

TAWM conditions estimation of the next state and reward on ∆t, as they depend on the temporal gap between the current and next state. The authors formulate M by modifying the world model of TD-MPC2 using 4-th order Runge-Kutta (RK4) method to enforce certain dynamical properties. Additionally, they modify the value model to take ∆t as an extra input. The authors train these models using various values of ∆t, which are log-uniformly sampled from a predefined interval.

**Other Comments Or Suggestions:**

More discussions about Generative World Model, especially video generative model-based (e.g., VideoAgent), are appreciated. Since Video generation directly learns the underlying dynamics. Note I am not requiring the authors to compare with VideoAgent, and I understand it’s different task setting.

**Other Strengths And Weaknesses:**

More theoretical insights (e.g., theorems) are appreciated.

**Questions For Authors:**

How can TAWM help reduce the sim2real gap? Is there any empirical or theoretical evidence?

**Relation To Broader Scientific Literature:**

This work belongs to the familiar of Model-based RL

**Theoretical Claims:**

No significant theorectial claims in my oppinion.

---

> ### Author Rebuttal · Authors · 2025-03-30
>
> Please see additional experimental results [here](https://sites.google.com/view/anonymous-site-rebuttal-6714).
>
> **Q1. Lack of theoretical claims**
>
> A1. To corroborate our empirical results, we offer additional theoretical analysis on the sample efficiency of our proposed time-aware world model. Here we provide only a brief overview of the analysis in this rebuttal; the full analysis will be included in the final version of our paper (we will provide proofs if requested in the next rebuttal).
>
> We start with following definitions:
>
> * $z(t)$: Function that gives state vector of the world at time step $t$. We assume this function is a locally Lipschitz function with a constant $L_1$. That is, for a sufficiently small $\Delta t > 0$, we assume $| z(t + \Delta t) - z(t) | \le L_1 \cdot \Delta t.$ This is a mild assumption, as we are only concerned about physical environments, where every physical property (e.g. position, velocity) changes continuously. Also, note that $L_1$ is proportional to the frequency of the underlying dynamics -- if the environment is almost static, $L_1$ would be near zero.
>
> * $a(t)$: Function that gives an action at time step $t$.
>
> * $f(z(t), a(t), \Delta t)$: Ground truth dynamics function that we want to approximate. This function satisfies the following equation for any pair of $z(t), a(t), $ and $\Delta t$: $z(t + \Delta t) = z(t) + f(z(t), a(t), \Delta t) \cdot \Delta t$.
>
> * $d(z(t), a(t), \Delta t)$: Dynamics function that we optimize to approximate $f$. We denote the predicted next state vector using this dynamics function as $\hat{z}(t + \Delta t) = z(t) + d(z(t), a(t), \Delta t) \cdot \Delta t$.
>
> Then, the following Lemma 1 holds.
>
> **Lemma 1**. For sufficiently small $0 < \Delta t_1 < \Delta t_2$, $| f(z(t), a(t), \Delta t_1) - f(z(t), a(t), \Delta t_2) \cdot \frac{\Delta t_2}{\Delta t_1} | < \frac{\Delta t_2 - \Delta t_1}{\Delta t_1} \cdot L_1$.
>
> Note that this relationship holds for every training data in our buffer -- therefore, we assume our dynamics function $d$ captures this relationship easily during training.
>
> **Assumption 1**. For sufficiently small $0 < \Delta t_1 < \Delta t_2$, $| d(z(t), a(t), \Delta t_1) - d(z(t), a(t), \Delta t_2) \cdot \frac{\Delta t_2}{\Delta t_1} | < \frac{\Delta t_2 - \Delta t_1}{\Delta t_1} \cdot L_2$.
>
> We can expect that $L_1$ would converge to $L_2$ during training. Based on these relationships, we can prove the following lemma.
>
> **Lemma 2**. For sufficiently small $0 < \Delta t_1 < \Delta t_2$, if $| f(z(t), a(t), \Delta t_2) - d(z(t), a(t), \Delta t_2) | = \epsilon$, we can compute the approximation error of the state vector as $| z(t + \Delta t_2) - \hat{z}(t + \Delta t_2) | = \epsilon \cdot \Delta t_2$. Then, for $\Delta t_1$, following holds: $| z(t + \Delta t_1) - \hat{z}(t + \Delta t_1) | \le \epsilon \cdot \Delta t_2 + (\Delta t_2 - \Delta t_1) \cdot (L_1 + L_2)$.
>
> This lemma tells us that when we decrease the approximation error $\epsilon$, it not only reduces the state approximation error at $(t + \Delta t_2)$, but also that of $(t + \Delta t_1)$. That is, when we optimize our model for one time step, it transfers to another time step. Also, it is more effective for the systems whose dynamics has lower frequency, and thus lower $L_1$. **Likewise, this lemma shows why our TAWM shows superior, or at least similar sample efficiency than the baseline, even though it has to learn additional temporal element.**
>
> --------------------------
>
> **Q2. Relationship to the video generative-model based world model (e.g. VideoAgent).**
>
> A2. Thank you for highlighting these works, which provide valuable insights that help connect our research with VLM Q&A. Both approaches share a common philosophy regarding the importance of temporal information. While VLM Q&A searches for the most relevant frames from pre-existing data to enhance video understanding, TAWM adaptively samples in time during the **data generation** process to capture the dynamics of underlying subsystems. In essence, VLM Q&A focuses on identifying key frames, whereas TAWM emphasizes aligning data generation with the system’s temporal dynamics. We will elaborate on this connection in the revised version.
>
> --------------------------
>
> **Q3. Possible application of TAWM for reducing sim2real gap.**
>
> A3. Although we don't yet have results on TAWM's learning transferability, we conjecture that TAWM can reduce the sim2real gap by adaptively sampling the frequency space to better synchronize **temporal effects** between simulated and real-world dynamics. This adaptive approach could enhance the robustness of the learning process by mitigating unexpected temporal noise from capturing devices or environmental factors—issues that are often absent in simulated environments. By training our world model across a wide range of frequencies, we expect it to become more resilient against these disturbances and improve its ability to generalize from simulation to real-world applications.

---

### Decision · Program_Chairs · 2025-05-01

**Decision:**

Accept (poster)

**Comment:**

This paper introduces the Time-Aware World Model (TAWM), a model-based reinforcement learning framework that conditions on variable time step sizes during training to improve generalization across environments with differing temporal dynamics. The reviewers acknowledged the method’s simplicity and empirical strength, highlighting its robustness and sample efficiency without increased training cost. Concerns were raised regarding the novelty of the approach, real-world applicability, and fairness of baseline comparisons. The rebuttal effectively addressed these points by providing theoretical analysis, ablation studies, and clarifications on the experimental setup and motivation. One reviewer raised their score following the response, citing stronger justification and statistical support. Given the clarity, empirical effectiveness, and practical utility of the approach, the overall assessment supports acceptance. Suggested directions for improvement include evaluation on real-world or sim-to-real tasks and broader comparisons with alternative world model approaches.